# Revealing the role of the human blood plasma proteome in obesity using genetic drivers

Shaza B. Zaghlool [1,11], Sapna Sharma[2,3,4,11], Megan Molnar [2,3], Pamela R. Matías-García[2,3,5], Mohamed A. Elhadad [2,3,6], Melanie Waldenberger [2,3,7], Annette Peters [3,4,7], Wolfgang Rathmann[4,8], Johannes Graumann [9,10], Christian Gieger[2,3,4], Harald Grallert[2,3,4,12] & Karsten Suhre [1,12✉]

Blood circulating proteins are confounded readouts of the biological processes that occur in different tissues and organs. Many proteins have been linked to complex disorders and are also under substantial genetic control. Here, we investigate the associations between over 1000 blood circulating proteins and body mass index (BMI) in three studies including over 4600 participants. We show that BMI is associated with widespread changes in the plasma proteome. We observe 152 replicated protein associations with BMI. 24 proteins also associate with a genome-wide polygenic score (GPS) for BMI. These proteins are involved in lipid metabolism and inflammatory pathways impacting clinically relevant pathways of adiposity. Mendelian randomization suggests a bi-directional causal relationship of BMI with LEPR/LEP, IGFBP1, and WFIKKN2, a protein-to-BMI relationship for AGER, DPT, and CTSA, and a BMI-to-protein relationship for another 21 proteins. Combined with animal model and tissue-specific gene expression data, our findings suggest potential therapeutic targets further elucidating the role of these proteins in obesity associated pathologies.

[1] Department of Physiology and Biophysics, Weill Cornell Medicine-Qatar, Doha, Qatar. [2] Research Unit of Molecular Epidemiology, Helmholtz Zentrum München, German Research Center for Environmental Health, Neuherberg, Bavaria, Germany. [3] Institute of Epidemiology, Helmholtz Zentrum München, German Research Center for Environmental Health, Neuherberg, Bavaria, Germany. [4] German Center for Diabetes Research (DZD), Neuherberg, Germany. [5] TUM School of Medicine, Technical University of Munich, Munich, Germany. [6] German Centre for Cardiovascular Research (DZHK), Partner Site Munich Heart Alliance, Munich, Germany. [7] German Research Center for Cardiovascular Research (DZHK), Partner Site Munich Heart Alliance, Munich, Germany. [8] Institute of Biometrics and Epidemiology, German Diabetes Center, Düsseldorf, Germany. [9] Scientific Service Group Biomolecular Mass Spectrometry, Max Planck Institute for Heart and Lung Research, W.G. Kerckhoff Institute, Bad Nauheim, Germany. [10] German Centre for Cardiovascular Research (DZHK), Partner Site Rhine-Main, Max Planck Institute of Heart and Lung Research, Bad Nauheim, Germany. [11] These authors contributed equally: Shaza B. Zaghlool, Sapna Sharma. [12] These authors jointly supervised this work: Harald Grallert, Karsten Suhre. ✉email: karsten@suhre.fr

O besity is a multifactorial disorder with still poorly understood causative mechanisms and a large polygenic contribution[1]. Genome-wide association studies of BMI identified genetic variants that can account for ~2.7–6% of the observed variance in body mass index (BMI)[2,3]. Due to an increasingly sedentary lifestyle and a transition to consumption of more and more processed foods, the prevalence of worldwide obesity has tripled over the past 40 years[4]. Based on the latest estimates in European Union countries, 30–70% of adults are affected by overweight and 10–30% by obesity (World Health Organization). Obesity greatly increases the risk of several chronic diseases such as depression, type 2 diabetes, cardiovascular disease, and certain cancers, putting a great burden on the healthcare system. Therefore, a better understanding of the interaction between lifestyle choices, environmental factors, and genetic predisposition is critical for developing effective treatments and preventive interventions[5,6].

The genetic composition is determined at conception and can be used to make predictions regarding disease susceptibility. The dramatic increase in obesity rates clearly points toward nongenetic factors or environmental factors as major drivers, most likely in interaction with genetic variants[7]. Although some diseases can result from a single rare monogenic mutation with a large effect, most common diseases are the consequence of a cumulative effect of polygenic inheritance encompassing numerous variants, each making only a small contribution to the overall disease risk[8]. Through genome-wide association studies (GWAS), more than 900 genetic variants have been identified to be associated with BMI[3]. However, these GWAS mapped associations still do not fully explain the molecular mechanisms leading to increased BMI. Genome-wide polygenic scores (GPS) are currently being used to quantify inherited disease susceptibility[9] and can explain ~13.9% of the variance in BMI which is more than twice the variance in BMI explained by using only the GWAS loci[3]. These scores approximated a normal distribution in the population and there is a considerable correlation between the GPS and measured BMI ($R^2 \sim 0.3$). Individuals in the upper tail of the GPS distribution can be susceptible to genetic effects comparable to carriers of single rare monogenic disease variants[9].

Given that proteins are the main building blocks of an organism, and also potential drug targets, proteome-wide association analysis seems to be the obvious next step in obesity research[10]. Levels of many proteins vary significantly between individuals with obesity and normal-weight individuals[11,12]. Until recently, mass spectrometry-based proteomic analyses of blood samples were limited to small sample sizes or a limited number of measured proteins. Multiplexed affinity-based proteomics approaches using antibodies or specifically designed aptamers now allow quantification of levels of hundreds of proteins from small amounts of plasma or serum samples. We previously quantified 1100 blood circulating proteins using the SOMAscan affinity proteomics platform (Somalogic Inc.)[13] in samples from 996 individuals of the population-based KORA F4 (Cooperative Health Research in the Region of Augsburg) study[14] and 356 participants of the multiethnic Qatar Metabolomics Study on Diabetes (QMDiab)[15].

In this work, we report a high throughput proteomics association study with BMI in KORA (Germany), and replication in two independent studies, including QMDiab, and publicly available association statistics from 3301 individuals in the INTERVAL study (England)[16]. We show that BMI is associated with several changes in plasma proteins. We compute GPS for BMI[17] and identify proteins whose levels associate with the GPS for BMI. We then use Mendelian randomization paired with experimental evidence to identify proteins and pathways that may

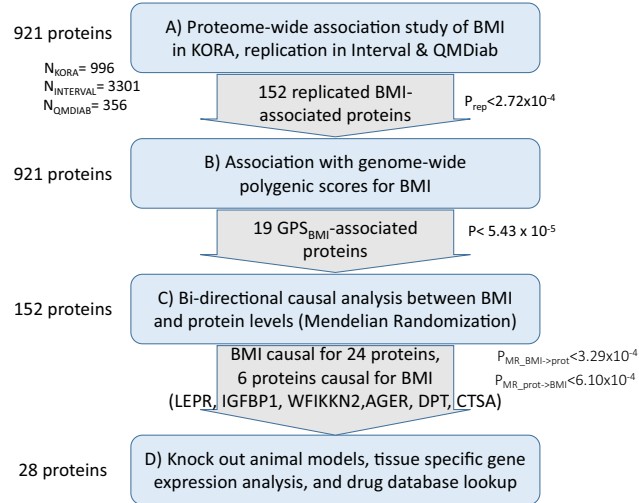

**Fig. 1 Study overview. A** Protein-wide association study with BMI conducted in KORA with confirmation/replication in INTERVAL and QMDiab. **B** Protein-wide association study with BMI polygenic scores (in KORA and QMDiab) **C** Bidirectional causality analysis to determine if BMI has a potentially causal effect on protein levels and/or if proteins are potentially causal in the development of obesity. **D** Studying tissue-specific gene expression in humans/mice, identification of genes encoding proteins related to obesity traits, searching for existing animal models, and identification of potentially targetable proteins through drug database searches.

be causally linked to obesity and vice versa. The study design and main findings are presented in Fig. 1.

## Results

**Out of 921, 152 assayed blood protein levels associate with BMI.** After stringent quality control, we identified 921 proteins whose levels were determined in 996 blood samples from the KORA study and that were also measured both in the INTERVAL and the QMDiab studies. Although not available for all three studies, we also included the leptin (LEP) and leptin receptor (LEPR) proteins for their well-studied roles in obesity. The study descriptive statistics for the 996 individuals are provided in Supplementary Data 1. We did not observe any significant differences ($p < 0.001$) in smoking and alcohol consumption between individuals with BMI ≥ 30 and BMI < 30. We used linear regression with age and sex as covariates to carry out a protein-wide association study in KORA and identified 184 associations between log2 transformed blood circulating protein levels and BMI after conservative Bonferroni correction ($p < 5.43 \times 10^{-5}$; 0.05/921). Totally, 107 proteins were negatively correlated with BMI while 77 were positively correlated (Fig. 2). The full summary statistics for the basic model (adjusting for age and sex only) are presented in Supplementary Data 2.

We tested the influence of a number of potential confounders on the BMI–protein associations, specifically, smoking status, alcohol consumption, physical activity, and diabetes state (Supplementary Data 3). We did not observe any substantial effect of confounding by these factors on the BMI–protein associations, and all Bonferroni significant protein–BMI associations found in the full model remained significant compared to the model where only age and sex were included as covariates.

We confirmed the BMI–protein associations using the published INTERVAL associations[16] and additionally attempted replication of these 184 associations in the multiethnic QMDiab study (Supplementary Data 4). Of the 184 proteins, 150 BMI–protein associations (81.5%) were replicated (both

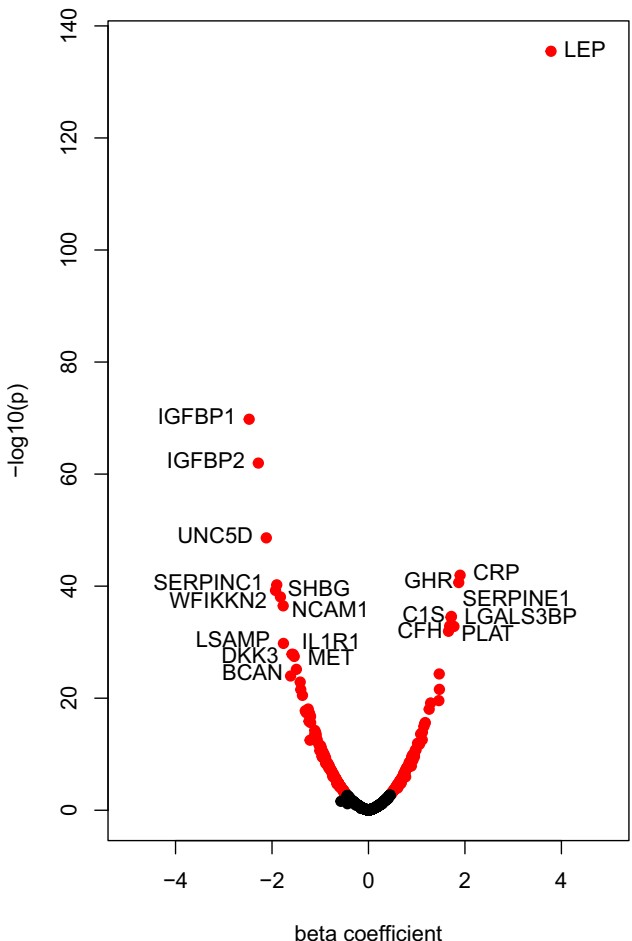

**Fig. 2 Protein-wide association study with body mass index.** Volcano plot showing the association of BMI with plasma protein levels in KORA using a linear regression model, including age and sex as covariates. Leptin is the strongest protein associated with BMI ($p = 3.34 \times 10^{-136}$) in addition to 151 significantly associated proteins (red).

significant and directionally concordant) in INTERVAL, after Bonferroni correction ($p < 2.72 \times 10^{-4}$; 0.05/184). In QMDiab, 37 (20.1%) of the BMI–protein associations were replicated after Bonferroni correction, while a further 131 proteins (71.2%) were directionally concordant, but were not sufficiently powered for replication. In total, 152 BMI–protein associations were replicated in at least one study—specifically, 35 associations replicated in both studies, 115 associations replicated only in INTERVAL, and two associations replicated only in QMDiab (THBS2 and ANGPT2). In addition, we found that out of 28 BMI–protein associations that had 95% replication power (determined by sampling), 17 proteins (60.7%) fully replicated in QMDiab, and 26 proteins (92.9%) displayed at least nominal significance. The Pearson correlation for the effect sizes is $R = 0.92$ between KORA and INTERVAL, and $R = 0.84$ between KORA and QMDiab (Supplementary Fig. 1).

**Association of BMI polygenic scores with BMI.** We computed GPS for BMI ($GPS_{BMI}$) for 996 participants of KORA and 353 of QMDiab (those with available genotyping data) using variants and weights from the previous studies[9,17]. Briefly, the $GPS_{BMI}$ score is based on summary statistics from a recent GWAS with BMI and assigns weights to each genetic variant depending on the strength of its association with BMI (see "Methods"). The $GPS_{BMI}$ was strongly associated with BMI in KORA ($p = 2.32 \times 10^{-43}$),

and was also significant in the multiethnic QMDiab study ($p = 5.54 \times 10^{-4}$) (Supplementary Fig. 2a, b).

Nineteen proteins were associated with the $GPS_{BMI}$ in KORA after accounting for multiple testing ($p < 5.43 \times 10^{-5}$; 0.05/921) (Supplementary Data 5). All 19 $GPS_{BMI}$-associated proteins were also strongly associated with BMI in KORA (Table 1). The regression coefficients for the BMI–protein associations and the $GPS_{BMI}$–protein associations were directionally concordant. The strongest protein association with BMI (LEP; $p = 3.34 \times 10^{-136}$) was also the strongest with $GPS_{BMI}$ ($p = 1.32 \times 10^{-12}$), followed by IGFBP1, IGFBP2, SERPINE1, and WFIKKN2.

We replicated the analysis in QMDiab to evaluate the applicability of a polygenic score derived from European participants to a cohort of mixed non-Caucasian ethnicity. Using linear regression and adjusting for age, sex, and study-specific covariates (described in "Methods"), five log2 transformed proteins remained significantly associated with $GPS_{BMI}$ after Bonferroni correction in QMDiab ($p < 0.05/19$; $2.63 \times 10^{-3}$) (NOTCH1, C5, NCAM1, CRP, and SERPINC1), while another six proteins (LEP, IGFBP1, WFIKKN2, UNC5D, MET, and RARRES2), were nominally associated with concordant directionality ($p < 0.05$) (Supplementary Data 6).

To confirm that the $GPS_{BMI}$ to protein associations were truly polygenic, as opposed to potentially being driven by a few strong in-cis variant effects, we excluded all genetic variants within 100 MB of the genes encoding the associated protein from score computation. All of the 19 protein associations with $GPS_{BMI}$ remained significant after eliminating potential cis-pQTL effects (Supplementary Data 7).

**Extreme BMI polygenic scores identify 19 proteomic signatures for 5% of the population.** Interestingly, the association between $GPS_{BMI}$ and BMI was not linear. The effect estimate was in fact much stronger at the extremes of the distribution (Fig. 3), which agrees with previous reports of this tail effect[9]. A tail effect is observed when the ratio of the effect at the tails to the effect of the entire distribution is greater than 1. To evaluate this tail effect in our study, we stratified the 996 KORA study samples based on $GPS_{BMI}$ percentiles. We found a steeper slope with respect to BMI and several protein measures (LEP, WFIKKN2, and IGFBP1) at the lower and upper extremes of the distribution. For instance, the mean BMI was 24.94 kg/m² [CI: 24.29–25.60] in the bottom decile and 31.09 kg/m² [CI: 31.09–32.21] in the top decile translating to a significant difference between the groups (two-sample $t$ test $p = 3.82 \times 10^{-17}$).

To investigate whether a similar tail effect can be observed for the associations between $GPS_{BMI}$ and blood circulating proteins, we compared the different effect sizes and significance levels at various percentiles of the $GPS_{BMI}$ distribution for all proteins (Supplementary Data 8), including the full dataset ($N = 996$), the 25th vs. 75th percentiles, the 20th vs. 80th percentiles, the 15th vs. 85th percentiles, the 10th vs. 90th percentiles, and the 5th vs. 95th percentiles. We found that the effect of $GPS_{BMI}$ on the log2 transformed LEP, IGFBP1, and WFIKKN2 was almost quadrupled in the 5% tail of the population compared to the full data (Fig. 4). Individuals in the extreme tail of the $GPS_{BMI}$ distribution showed an over-proportionally increased genetic predisposition for developing obesity[17]. We found a similar tail effect for all 19 $GPS_{BMI}$-associated proteins (Table 2) (Supplementary Fig. 3).

We further tested whether a similar tail effect was observed for the remaining 133 BMI-associated proteins (Supplementary Data 9). Out of 133, 81 proteins were associated with BMI ($p < 5.43 \times 10^{-5}$), but weakly associated with $GPS_{BMI}$ ($0.05 \leq p \leq 6.30 \times 10^{-5}$). Of these 81 proteins, 49 proteins were weakly associated with the tails of $GPS_{BMI}$ ($p < 0.05$) and had a greater

**Table 1 Proteins significantly associated with BMI and the polygenic score for BMI.** The $p$ values ($p_{BMI-prot}$), linear regression coefficients ($b_{BMI-prot}$) are for the BMI-protein associations, while $p_{GPS-prot}$ and $b_{GPS-prot}$ are for the $GPS_{BMI}$–protein linear regression associations.

| Protein Soma SeqID (Entrez Gene) | $p_{BMI-prot}$ | $b_{BMI-prot}$ | $SE_{BMI}$ | $p_{GPS-prot}$ | $b_{GPS-prot}$ | $SE_{GPS}$ |
|---|---|---|---|---|---|---|
| Leptin 2575-5_5 (LEP) | $3.34 \times 10^{-136}$ | 0.122 | 0.004 | $1.32 \times 10^{-12}$ | 0.181 | 0.025 |
| Insulin-like growth factor-binding protein 1 2771-35_2 (IGFBP1) | $1.60 \times 10^{-70}$ | −0.110 | 0.006 | $4.72 \times 10^{-10}$ | −0.187 | 0.030 |
| Insulin-like growth factor-binding protein 2 2570-72_5 (IGFBP2) | $1.08 \times 10^{-62}$ | −0.107 | 0.006 | $3.60 \times 10^{-9}$ | −0.183 | 0.031 |
| Plasminogen activator inhibitor 1 2925-9_1 (SERPINE1) | $3.53 \times 10^{-35}$ | 0.083 | 0.006 | $2.84 \times 10^{-8}$ | 0.175 | 0.031 |
| WAP, Kazal, immunoglobulin, Kunitz, and NTR domain-containing protein 2 3235-50_2 (WFIKKN2) | $8.30 \times 10^{-39}$ | −0.086 | 0.006 | $5.83 \times 10^{-8}$ | −0.168 | 0.031 |
| Dickkopf-related protein 3 3607-71_1 (DKK3) | $1.40 \times 10^{-28}$ | −0.074 | 0.006 | $7.85 \times 10^{-7}$ | −0.152 | 0.031 |
| Galectin-3-binding protein 5000-52_1 (LGALS3BP) | $2.53 \times 10^{-35}$ | 0.083 | 0.006 | $1.18 \times 10^{-6}$ | 0.153 | 0.031 |
| Sex hormone-binding globulin 4929-55_1 (SHBG) | $5.84 \times 10^{-40}$ | −0.084 | 0.006 | $1.88 \times 10^{-6}$ | −0.142 | 0.030 |
| Growth hormone receptor 2948-58_2 (GHR) | $2.28 \times 10^{-41}$ | 0.089 | 0.006 | $3.38 \times 10^{-6}$ | 0.145 | 0.031 |
| Growth/differentiation factor 2 4880-21_1 (GDF2) | $3.13 \times 10^{-21}$ | −0.063 | 0.007 | $4.35 \times 10^{-6}$ | −0.141 | 0.031 |
| Netrin receptor UNC5D 5140-56_3 (UNC5D) | $2.43 \times 10^{-49}$ | −0.093 | 0.006 | $4.63 \times 10^{-6}$ | −0.137 | 0.030 |
| Neurogenic locus notch homolog protein 1 5107-7_2 (NOTCH1) | $1.31 \times 10^{-23}$ | −0.068 | 0.007 | $5.08 \times 10^{-6}$ | −0.143 | 0.031 |
| Hepatocyte growth factor receptor 2837-3_2 (MET) | $3.52 \times 10^{-28}$ | −0.075 | 0.007 | $6.68 \times 10^{-6}$ | −0.142 | 0.031 |
| Antithrombin-III 3344-60_4 (SERPINC1) | $5.85 \times 10^{-41}$ | −0.087 | 0.006 | $6.82 \times 10^{-6}$ | −0.138 | 0.030 |
| C-reactive protein 4337-49_2 (CRP) | $1.10 \times 10^{-42}$ | 0.091 | 0.006 | $2.99 \times 10^{-5}$ | 0.131 | 0.031 |
| Neural cell adhesion molecule 1, 120 kDa isoform, 4498-62_2 (NCAM1) | $3.31 \times 10^{-37}$ | −0.085 | 0.006 | $3.05 \times 10^{-5}$ | −0.131 | 0.031 |
| Protein jagged-1 5092-51_3 (JAG1) | $2.47 \times 10^{-7}$ | −0.036 | 0.007 | $3.45 \times 10^{-5}$ | −0.130 | 0.031 |
| Cystatin-M 3303-23_2 (CST6) | $2.24 \times 10^{-11}$ | −0.044 | 0.006 | $3.56 \times 10^{-5}$ | −0.123 | 0.030 |
| Endothelial cell-specific molecule 1 3805-16_2 (ESM1) | $4.92 \times 10^{-18}$ | −0.059 | 0.007 | $4.33 \times 10^{-5}$ | −0.129 | 0.031 |

than 3-fold increase/decrease in effect size between the 5% tails and the full data. On the other hand, 52 out of 133 proteins were associated with BMI, but not at all with $GPS_{BMI}$ ($p > 0.05$). However, 11 of these 52 proteins were associated with the tails of $GPS_{BMI}$ ($p < 0.05$) and had a greater than 3-fold increase/decrease in effect size between the 5% tails and the full data.

**Mendelian randomization.** To assess whether proteins are causally affected by BMI in the direction (BMI-to-protein) or vice versa (protein-to-BMI), we carried out bi-directional Mendelian randomization investigations. We initially conducted both, a one-sample (1SMR) and a two-sample (2SMR) Mendelian randomization analysis, and in both directions (Table 3). MR analysis results are presented using the 2SLS method for the 1SMR, and using the IVW method for the 2SMR. In the BMI-to-protein direction, we used $GPS_{BMI}$ as an instrument for BMI. Our results indicated that the 1SMR had higher statistical power than the 2SMR in identifying significant MR associations. This is plausible because the BMI instrument was generated using variant weights from the largest GWAS with BMI. The 1SMR used individual-level protein data, while the 2SMR only had access to protein summary statistics from a study that is merely four times the size of KORA.

In the protein-to-BMI direction, we found that the 2SMR was more powered than the 1SMR. This was also plausible and could be attributed to the fact that individual-level genetic associations with BMI as an outcome, were much weaker in a study the size of KORA, while the effect estimates from larger GWAS with BMI[2,3] were much more precise. In all applicable cases (including nominal associations), we found consistency in the MR effect directions between the 1SMR and 2SMR, and in both directions of the MR (BMI-to-protein, and protein-to-BMI) (Supplementary Data 10–13). We, therefore, focus our analysis on 1SMR in the BMI-to-protein direction and 2SMR in the protein-to-BMI direction.

**BMI is potentially causal for 24 of the 152 tested proteins.** The 1SMR approach allowed the investigation of potentially causal relationships between BMI and 152 replicated blood plasma proteins. Our analysis suggests that BMI has a causal effect on 24 proteins, after correction for multiple testing ($p < 0.05/152 = 3.29 \times 10^{-4}$) (Fig. 5, Supplementary Data 10).

**Six plasma proteins have a potentially causal role in the development of obesity.** Using 2SMR analysis, we used the Proteome PheWAS browser[18] which curated single-nucleotide

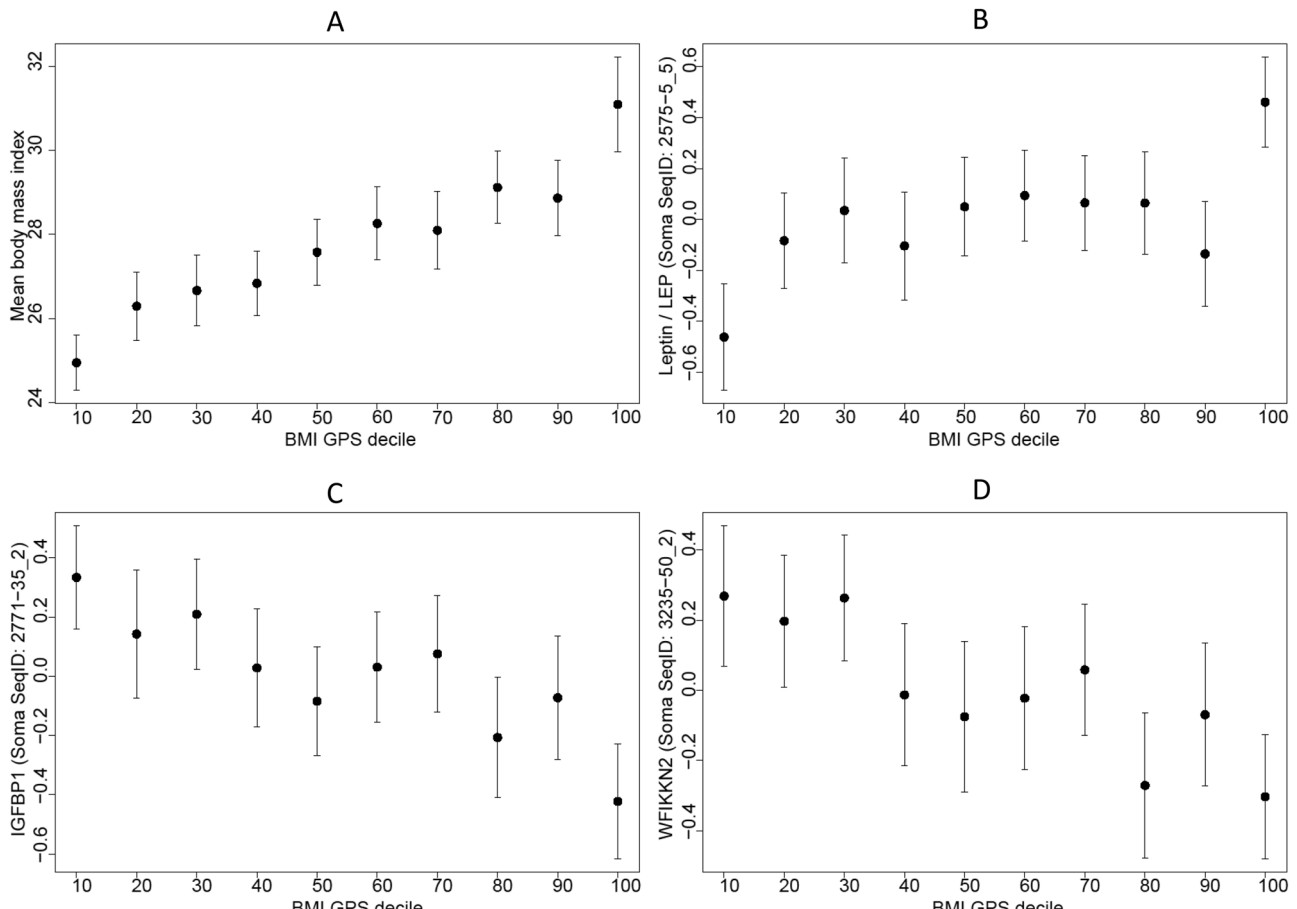

**Fig. 3 Stratification of the KORA samples according to GPS$_{BMI}$ deciles ($n = 996$ biologically independent samples).** There is a steep slope with respect to both BMI (**A**) and various protein measures (**B**–**D**) at the upper and lower deciles. LEP, like BMI, has an increasing trend, while IGFBP1 and WFIKKN2 has a decreasing trend. The centers are the mean protein values and the error bars are the 95% confidence intervals.

polymorphisms (SNPs) associated with proteins from five protein GWASs[13,16,19–21] and categorized protein instruments based on their suitability for MR analysis. We identified genetic instruments for 82 of the 152 replicated proteins, in addition to LEPR which we considered a positive control in this study. This analysis suggested that six proteins (LEPR, IGFBP1, WFIKKN2, AGER, DPT, and CTSA) may potentially have a causal role in the development of obesity, after correction for multiple testing ($p < 0.05/82 = 6.10 \times 10^{-4}$) (Supplementary Data 13).

In summary, we found that BMI had a causal effect on the levels of 24 proteins, while six proteins had a potentially causal role in the development of obesity, two of which are suggested to have roles in both directions (IGFBP1 and WFIKKN2). LEPR/LEP also showed roles in both directions (LEPR-to-BMI, and BMI-to-LEP).

**The biological role of the 28 causally and/or consequentially BMI-associated proteins.** To study the tissue-specific role of the causal/consequential proteins in obesity, we screened the Genotype-Tissue Expression (GTEx) human database and the Mouse Genome Informatics (MGI) database. We found that data from GTEx was available for 20 of the 28 proteins and those proteins can be clustered into two groups (Supplementary Fig. 4A). While the functional role of a subset of these may be more global, others may imply specific pathways. The first cluster consisting of seven proteins showed wide-spread expression across 54 different human tissues and similarly across various

mouse tissues. For instance, NCAM1, DKK3, IGFBP2, LGALS3BP, SERPINE1, and NOTCH1 were predominantly expressed in the brain, adipose, and heart tissues. The second cluster consisting of 13 proteins, showed more sporadic expression in relevant human tissues, such as LEP in adipose tissue, IGFBP1, CRP, and SERPINC1 in liver tissue, CST6 in the skin, and WFIKKN2 in ovaries, testis, and brain. In mice, Wfikkn2 is primarily expressed in the brain, heart, eyes, and pancreas (Supplementary Fig. 4B). There is not much knowledge about the role of WFIKKN2 in obesity, however, it is known to have a regulatory role of some members of the transforming growth factor-beta (TGFB) family[22]. The TGFB superfamily is produced in adipose tissues and involved in the regulation of adiposity[23] and obesity is known to alter their expression level.

We further investigated the 28 proteins for enrichment in obesity traits, using the hybrid mouse diversity panel (HMDP)[24] as well as an F2 cross of the inbred ApoE−/− C57BL/6J and C3H/HeJ strains[25] (see "Methods"). Twenty-six of the proteins had mouse orthologs[26]. We observed correlations ($R > 0.1$ and $p$ value $< 0.05$) between adipose, liver, and brain tissue gene expression and numerous essential obesity traits, including body fat composition, bone density, insulin, and various lipid traits such as LDL, HDL, cholesterol, triglycerides, etc. (Fig. 6). We observed enrichment in obesity traits in mice, such as weight, length, and triglycerides for the potentially causal proteins (Ager, Ctsa, and Dpt). We also found enrichment in HDL cholesterol and total cholesterol for Ager and Ctsa, fat mass for Ctsa and Dpt, and abdominal fat for Dpt. In general, correlations between

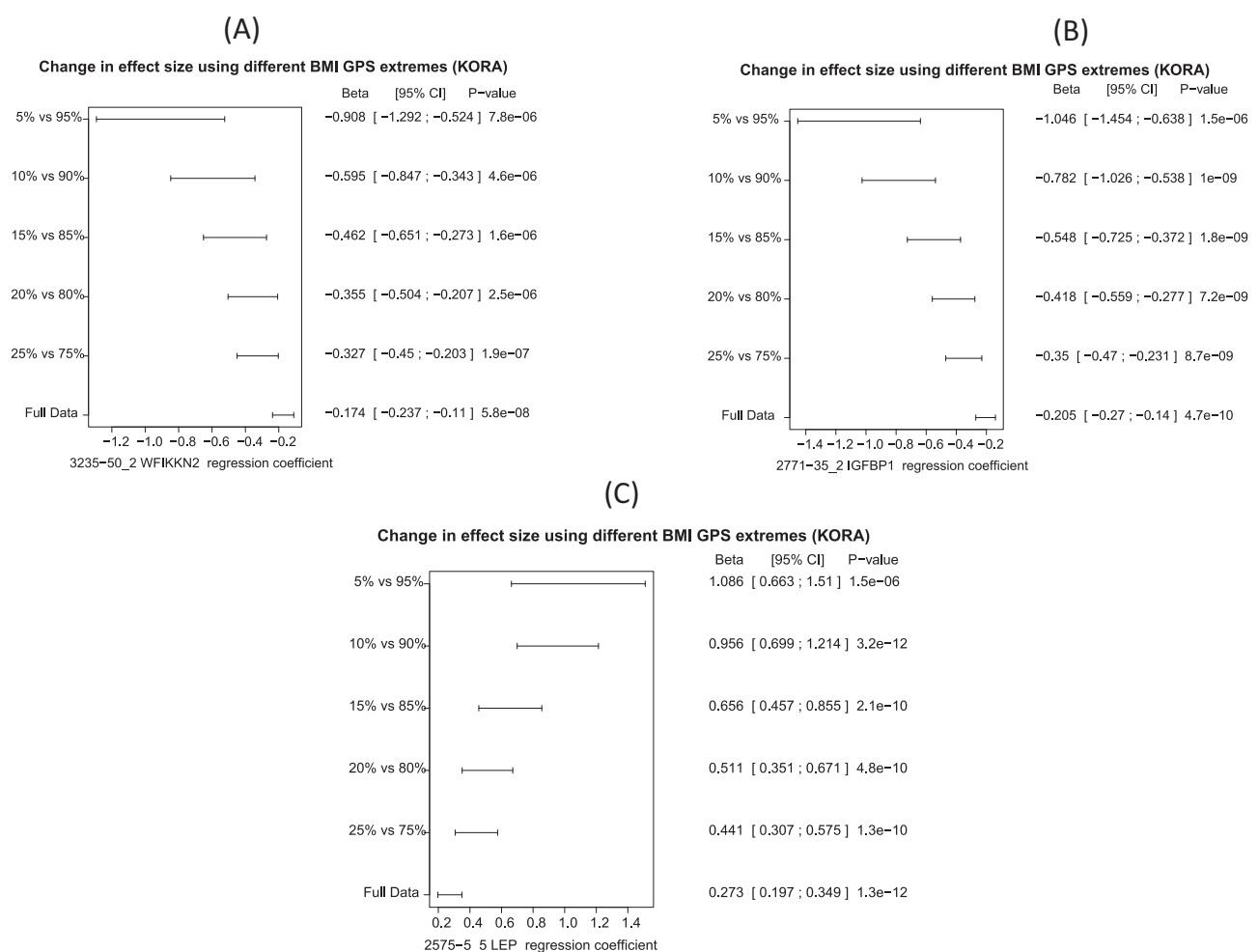

**Fig. 4 Extreme GPS_BMI is a strong risk factor for increased protein levels and increased BMI ($n = 996$ biologically independent samples).** The effects from the linear regression model of GPS_BMI on **A** WFIKKN2, **B** IGFBP1, and **C** LEP are almost quadrupled in the extreme 5% of the sample compared to the full data ($n = 996$). The centers are the regression coefficients (betas) and the error bars are the 95% confidence intervals.

proteins were similar, but stronger in adipose tissue compared to liver and brain tissues. However, a number of differences are noteworthy. For instance, Wfikkn2 was associated with triglycerides and lipids in adipose and brain tissue, but not the liver. The association of Wfikkn2 and triglycerides and total fat in brain tissue was previously reported[27].

We then queried Phenoscanner[28,29] to determine which of the 152 BMI-associated proteins were associated with known BMI loci or may be considered the best candidate in the genomic vicinity. We found that five proteins were strong pQTLs for BMI loci/regions. These included leptin (LEP), C-reactive protein (CRP), apolipoprotein B (APOB), lysosomal protective protein (CTSA), and neural cell adhesion molecule 1, 120 kDa isoform (NCAM1). Further proteins were found to represent eQTLs near BMI loci, including immunoglobulin M (IGJ), interleukin-1 receptor accessory protein (IL1RAP), calpastatin (CAST), apolipoprotein B (APOB), platelet-activating factor acetylhydrolase (PLA2G7), plasma protease C1 inhibitor (SERPING1), reticulon-4 receptor (RTN4R), insulin-like growth factor 1 receptor (IGF1R), integrin alpha-V: beta-5 complex (ITGB5), complement factor B (CFB), complement component 1 Q subcomponent-binding protein, mitochondrial (C1QBP), a cell adhesion molecule 1 (CADM1), galectin-3-binding protein (LGALS3BP), and antithrombin-III (SERPINC1).

Lastly, and in order to identify drug targets for the potential treatment of obesity, we used the DrugBank database[30] to search for existing drugs that target the six proteins that were causal for BMI. Three proteins were targets for at least one existing drug that has completed phase II clinical trials (Supplementary Data 15). Drugs likely for treating obesity included Metreleptin[31], which targets leptin receptors, to treat complications of leptin deficiency in individuals with congenital or acquired lipodystrophy. Another drug, Pegvisomant[32], is a highly selective growth hormone (GH) receptor antagonist that is used to treat acromegaly by the production of IGF-1 which is the main mediator of GH activity. A third drug, Mecasermin[33], targets IGFBP1 and IGFBP2 by acting as an agonist of insulin-like growth factor 1 receptor. It is a drug that is used for the treatment of growth failure in pediatric patients with primary IGFD or GH gene deletion. Although the latter drug actually exacerbates growth, the affected pathway(s) may still be considered a potential target for medical intervention.

## Discussion

**Proteins associated with obesity and obesity score.** Blood circulating proteins permeate the entire organism and may be involved in the direct regulation of complex diseases such as obesity or diabetes. Protein associations may provide biological interpretations of the molecular mechanisms occurring due to increased BMI and obesity. We identified 152 proteins that were significantly associated with BMI in the KORA study and

**Table 2 The over-proportional contribution of genetics to BMI in the tail of the GPS$_{BMI}$ distribution translates to at least a threefold increase/decrease in protein levels. The effect sizes (beta) and $p$ values from the linear regression models are presented for the full data set and limited to data in the 5% tails of the GPS$_{BMI}$, respectively.**

| Protein | Beta$_{full}$ | $p_{full}$ | Beta$_{5\%}$ | $p_{5\%}$ | $\frac{Beta_{5\%}}{Beta_{full}}$ |
|---|---|---|---|---|---|
| IGFBP2 | −0.189 | $3.60 \times 10^{-9}$ | −1.073 | $1.08 \times 10^{-7}$ | 5.671 |
| IGFBP1 | −0.205 | $4.72 \times 10^{-10}$ | −1.046 | $1.49 \times 10^{-6}$ | 5.101 |
| LEP | 0.273 | $1.32 \times 10^{-12}$ | 1.086 | $1.52 \times 10^{-6}$ | 3.977 |
| SERPINE1 | 0.175 | $2.84 \times 10^{-8}$ | 1.002 | $4.07 \times 10^{-6}$ | 5.732 |
| UNC5D | −0.152 | $4.63 \times 10^{-6}$ | −1.102 | $6.89 \times 10^{-6}$ | 7.230 |
| WFIKKN2 | −0.174 | $5.83 \times 10^{-8}$ | −0.908 | $7.83 \times 10^{-6}$ | 5.227 |
| SHBG | −0.159 | $1.88 \times 10^{-6}$ | −0.958 | $1.43 \times 10^{-5}$ | 6.014 |
| NCAM1 | −0.132 | $3.05 \times 10^{-5}$ | −0.723 | $1.85 \times 10^{-4}$ | 5.464 |
| NOTCH1 | −0.145 | $5.08 \times 10^{-6}$ | −0.728 | $2.16 \times 10^{-4}$ | 5.011 |
| MET | −0.143 | $6.68 \times 10^{-6}$ | −0.718 | $2.27 \times 10^{-4}$ | 5.034 |
| DKK3 | −0.160 | $7.85 \times 10^{-7}$ | −0.857 | $2.67 \times 10^{-4}$ | 5.360 |
| SERPINC1 | −0.147 | $6.82 \times 10^{-6}$ | −0.793 | $7.65 \times 10^{-4}$ | 5.408 |
| CST6 | −0.139 | $3.56 \times 10^{-5}$ | −0.818 | $8.66 \times 10^{-4}$ | 5.894 |
| JAG1 | −0.132 | $3.45 \times 10^{-5}$ | −0.701 | $1.35 \times 10^{-3}$ | 5.294 |
| LGALS3BP | 0.154 | $1.18 \times 10^{-6}$ | 0.715 | $1.69 \times 10^{-3}$ | 4.651 |
| GDF2 | −0.149 | $4.35 \times 10^{-6}$ | −0.676 | $4.40 \times 10^{-3}$ | 4.524 |
| GHR | 0.148 | $3.38 \times 10^{-6}$ | 0.570 | $7.62 \times 10^{-3}$ | 3.841 |
| ESM1 | −0.130 | $4.33 \times 10^{-5}$ | −0.514 | $2.00 \times 10^{-2}$ | 3.952 |
| CRP | 0.133 | $2.99 \times 10^{-5}$ | 0.406 | $5.11 \times 10^{-2}$ | 3.044 |

and 2 (IGFBP1 and IGFBP2), and GH have central pathophysiological roles in normal metabolism, insulin resistance, and type 2 diabetes[34], mainly by modulating insulin sensitivity. IGFBP2 has been shown to be regulated by leptin[35]. GH also has an important role in the development of obesity, and GH receptor (GHR) mutations were reported in individuals with obesity[36]. Netrin receptor (UNC5D), is a mediator of inflammation known to promote macrophage retention in adipose tissue[37]. Markers of high adiposity also include increased concentrations of CRP[38,39] and SERPINE1[40], contrasted by reduced levels of SERPINC1[41] and SHBG[42], all of which we observe in this study.

Common diseases are a result of a complex interplay between genetics and a broad range of environmental perturbations. Exposure to environmental factors (i.e., diet, age, exposure to toxins) activates highly interacting protein networks[43], which in turn, may drive molecular mechanisms toward disease. This is likely the case for obesity, where environmental contributions to BMI are well recognized[19]. A tail effect similar to the one previously reported for GPS$_{BMI}$[17] was observed for the 19 GPS associated proteins and another 49 BMI-associated proteins (at a weaker threshold), but not for 52 proteins that were associated with BMI and not with GPS$_{BMI}$. The former set of proteins supports the presence of a strong genetic effect on the proteins. Thus, the driving factors for obesity may be distinguished as genetic or other environmental factors. This may aid in explaining clinical observations, such as why a subset of individuals experiences an earlier onset of obesity and may be useful in defining treatment strategies.

confirmed them in at least one other study. We then applied a GPS that was derived and validated in a previous study[17] to compute a GPS$_{BMI}$ for nearly 1000 individuals from the KORA study. The genetic background of the KORA participants is similar to the cohort on which the score computation was based, that is, both are of European ancestry[2]. The GPS$_{BMI}$ was strongly associated with BMI, and also with differences in leptin levels, a protein whose association with BMI and obesity is well established, and numerous other obesity-related proteins including WFIKKN2 and IGFBP1. The GPS$_{BMI}$ not only captures strong BMI variants but genome-wide BMI effects (although the former would have stronger weights). The overlap we found between BMI loci and pQTLs/eQTLs ie. LEP, CRP, NCAM1, CTSA, LGALS3BP, IGJ, and SERPINC1 provides useful insight for causation. We later used MR to distinguish between the contribution/consequence of these proteins with respect to BMI.

In this study, we test a genetic score that was based on European ancestry in a population of Arabs and mixed ethnicities. Despite the fact that QMDiab has a linkage disequilibrium (LD) structure differing from European populations, as well as being multiethnic, diabetes-directed, and of limited power, we none the less observe an association, supporting the robustness and strength of the observed signals. As we used diabetes as a covariate, signals are likely driven by BMI rather than diabetes. Thus, while the association between the GPS$_{BMI}$ and BMI in QMDiab compared to the European cohort may have been weaker, the signals we do replicate are likely strong true positives. It remains open to speculation whether generalization of the GPS$_{BMI}$ score was limited due to differences in the genetic architecture of obesity between the populations or due to the sample size.

The 19 proteins that were associated with GPS$_{BMI}$ displayed an amplified effect for the individuals at the tail of the population distribution. These proteins all play important roles in obesity and have been well documented in the literature. For instance, proteins of the insulin-like growth factor system and their receptors including insulin-like growth factor-binding protein 1

**Causal analysis.** Our study suggests that BMI has a potentially causal effect on 24 proteins. Causality in MR is defined as the fact that a modification of exposure leads to a change in the outcome. Causality in this sense is not indicative of a particular molecular mechanism per se. It simply suggests that modifying the exposure will necessarily lead to a predictable effect on the outcome. Our observation is in line with a previous study that reports widespread effects of adiposity on DNA methylation[7]. On the other hand, our 2SMR also suggests that LEPR, IGFBP1, WFIKKN2, AGER, DPT, and CTSA are potentially causal for the development of obesity. Taken together, our data suggest that a bidirectional relationship is likely and may be replicated in other BMI-protein associations due to the underlying complexity of the disease and the multitude of involved pathways.

**Animal models.** We extensively searched the literature for animal models for the 28 proteins (Supplementary Data 14) and found models covering 18 out of 28 protein-coding genes linked with obesity. Mice studies showed that Lepr knockout mice became excessively obese[44]. Lep and Lepr deficient mice have also been shown to be hyperinsulinemic, hyperglycemic (depending on the age and strain), and have elevated total cholesterol levels and LDL/HDL1 particles[45–47]. Lep and Lepr levels may be both a sensor of fat mass and at the same time, part of a negative feedback mechanism to maintain a set point for body fat storage[48]. In some instances, such as leptin deficiency in monogenic obesity, the causal role of Lep on BMI is obvious[49]. A global Igfbp1 deletion in mice showed a significant increase in body weight and body fat mass[50]. Interestingly, epigenetic regulation of IGFBP2 has also been suggested to play a role in abdominal obesity[51].

WAP, Kazal, immunoglobulin, Kunitz, and NTR domain-containing protein 2 (WFIKKN2) is a protease-inhibitor that contains multiple distinct protease inhibitor domains[52]. WFIKKN2 encodes growth and differentiation factor-associated serum protein-1 (GASP1)[53] and WFIKKN2 is an inhibitory

**Table 3 Accumulative evidence is suggestive of relationships between BMI and proteins in both directions. MR analysis is summarized for both directions (BMI-to-protein and protein-to-BMI) for one-sample MR (1SMR) using the 2SLS method (linear regression), and two-sample MR (2SMR) using the IVW method. Entries with an asterisk are Bonferroni significant (corrected for the respective number of MR tests).**

| Direction | BMI→Protein | | | | Protein→BMI | | | | Animal model |
|---|---|---|---|---|---|---|---|---|---|
| Method | 1SMR (KORA) | | 2SMR (GIANT + UKBiobank/INTERVAL) | | 1SMR (KORA) | | 2SMR (PheWeb browser) | | |
| Protein | Beta | p | Beta | p | Beta | p | Beta | p | |
| LEP (2575-5_5) | 0.104 | $1.43 \times 10^{-8}$* | 0.424 | $1.01 \times 10^{-6}$* | No instruments | | No instruments | | Yes |
| LEPR (5400-52_3) | Not measured in KORA | | −0.212 | $1.14 \times 10^{-2}$ | Not measured in KORA | | 0.007 | $4.62 \times 10^{-3}$* | Yes |
| IGFBP1 (2771-35_2) | −0.123 | $7.47 \times 10^{-12}$* | −0.225 | $8.05 \times 10^{-3}$ | No instruments | | −0.053 | $5.51 \times 10^{-4}$* | Yes |
| IGFBP2 (2570-72_5) | −0.121 | $9.73 \times 10^{-11}$* | −0.277 | $9.48 \times 10^{-4}$* | No instruments | | No instruments | | Yes |
| SERPINE1 (2925-9_1) | 0.109 | $3.04 \times 10^{-8}$* | 0.208 | $1.62 \times 10^{-2}$ | No instruments | | No instruments | | No |
| WFIKKN2 (3235-50_2) | −0.109 | $2.49 \times 10^{-8}$* | −0.290 | $5.00 \times 10^{-4}$* | −0.134 | 0.749 | −0.016 | $7.39 \times 10^{-5}$* | Yes |
| DKK3 (3607-71_1) | −0.102 | $4.70 \times 10^{-7}$* | −0.235 | $5.19 \times 10^{-3}$ | −1.226 | 0.064 | −0.009 | 0.246 | No |
| LGALS3BP (5000-52_1) | 0.092 | $2.24 \times 10^{-6}$* | Data not available | | No instruments | | 0.018 | 0.279 | Yes |
| SHBG (4929-55_1) | −0.091 | $1.85 \times 10^{-6}$* | Data not available | | No instruments | | 0.014 | 0.233 | Yes |
| GHR (2948-58_2) | 0.095 | $1.18 \times 10^{-6}$* | 0.212 | $1.11 \times 10^{-2}$ | No instruments | | No instruments | | Yes |
| GDF2 (4880-21_1) | −0.094 | $4.12 \times 10^{-6}$* | Data not available | | No instruments | | No instruments | | Yes |
| UNC5D (5140-56_3) | −0.092 | $1.08 \times 10^{-6}$* | −0.197 | $2.25 \times 10^{-2}$ | No instruments | | −0.001 | 0.978 | No |
| NOTCH1 (5107-7_2) | −0.091 | $4.98 \times 10^{-6}$* | −0.233 | $5.44 \times 10^{-3}$ | −0.257 | 0.758 | No instruments | | Yes |
| MET (2837-3_2) | −0.085 | $1.36 \times 10^{-5}$* | −0.087 | 0.296 | 0.002 | 0.997 | No instruments | | Yes |
| SERPINC1 (3344-60_4) | −0.081 | $2.60 \times 10^{-5}$* | −0.057 | 0.491 | No instruments | | No instruments | | No |
| CRP (4337-49_2) | 0.074 | $1.25 \times 10^{-4}$* | 0.314 | $2.40 \times 10^{-4}$* | No instruments | | −0.012 | 0.205 | No |
| NCAM1 (4498-62_2) | −0.082 | $2.43 \times 10^{-5}$* | −0.304 | $3.25 \times 10^{-4}$* | No instruments | | 0.015 | 0.197 | No |
| JAG1 (5092-51_3) | −0.086 | $5.40 \times 10^{-5}$* | −0.195 | $2.22 \times 10^{-2}$ | −0.312 | 0.818 | −0.011 | 0.321 | No |
| CST6 (3303-23_2) | −0.071 | $6.56 \times 10^{-4}$* | −0.084 | 0.320 | No instruments | | No instruments | | No |
| ESM1 (3805-16_2) | −0.086 | $3.27 \times 10^{-5}$* | −0.255 | $2.33 \times 10^{-3}$ | No instruments | | No instruments | | No |
| ADIPOQ (3554-24_1) | −0.078 | $9.14 \times 10^{-5}$* | Data not available | | No instruments | | No instruments | | Yes |
| IBSP (3415-61_2) | −0.079 | $1.13 \times 10^{-4}$* | No instruments | | No instruments | | No instruments | | No |
| C1S (3590-8_3) | 0.086 | $9.67 \times 10^{-6}$* | Data not available | | 0.869 | 0.028 | No instruments | | No |
| POSTN (3457-57_1) | −0.081 | $1.87 \times 10^{-4}$* | No instruments | | −1.519 | 0.077 | No instruments | | Yes |
| IGHM (3069-52_3) | −0.078 | $2.09 \times 10^{-4}$* | Data not available | | No instruments | | No instruments | | No |
| AGER (4125-52_2) | −0.030 | 0.141 | −0.189 | $2.70 \times 10^{-2}$ | No instruments | | −0.041 | $6.09 \times 10^{-5}$* | Yes |
| DPT (4979-34_2) | 0.019 | 0.367 | 0.109 | 0.192 | −1.519 | 0.114 | −0.023 | $1.24 \times 10^{-4}$* | Yes |
| CTSA (3179-51_2) | 0.036 | $8.40 \times 10^{-2}$ | 0.209 | $1.70 \times 10^{-2}$ | 0.672 | 0.320 | −0.075 | $1.18 \times 10^{-7}$* | Yes |

TGF-β binding protein. Animal knockout models for Wfkinn2/Gasp1 have also shown a significant increase in body weight, particularly in muscle mass[54].

Further, Ager knockout models showed weight gain and increased plasma cholesterol[55], and Ctsa knockout mice presented with thick skin that contained enlarged hyperplastic epidermal glands as well as a reduction in dermal fat[56]. The circulating soluble receptor for advanced glycation end products (AGER) is negatively associated with BMI[57], as we also observed in KORA. In addition, recent evidence suggests a role of adipokine dermatopontin (DPT) in obesity by regulation of adipose tissue remodeling and inflammation[58]. A Dpt knockout mouse model showed increased subcutaneous adipose tissue[59], and effects on skin elasticity, dermis thickness, and collagen accumulation.

**Other biological evidence**. GASP1/WFKINN2 has mainly been involved in skeletal and muscle fiber development in the heart[60]. Higher WFIKKN2 protein levels were associated with lower levels of fasting insulin, triglycerides, HOMA-IR, and visceral fat[61] suggesting a protective role against metabolic dysregulation.

In addition, global overexpression of Wfikkn2/Gasp1 resulted in a significant increase in body weight in mice[54]. Interestingly, our findings were consistent with a recent SOMAscan protein study of type 2 diabetes in AGES-Reykjavik, where WFIKKN2 was reported to be potentially causal for type 2 diabetes[62],

independent from BMI. Furthermore, WFIKKN2 was suggested as a potentially causal candidate for type 2 diabetes, in a second study from INTERVAL that associates the diabetes risk score with proteins[63], after adjusting for age, sex, and technical covariates. Lastly, genetic variants in the WFIKKN2 locus (cis-pQTLs) showed regulation of GDF8/11 at the protein level in a trans-pQTL manner[16]. Thus, the plasma levels of GDF8/11 and WFIKKN2 are strongly controlled by genetics. Genetically supported targets could be more successful than those without genetic support in clinical practice[64], suggesting that WFIKKN2 is a potential target that would modulate GDF8/11 function, as suggested in Sun et al.[16].

**Limitations**. We are aware of several limitations to our study. First, the SOMAscan technology provides a relative abundance of protein levels, not absolute concentrations. However, this is not a concern for association studies. Second, the findings reported here are limited to the specific protein set targeted by the SOMAscan panel, and also to protein associations that are detectable in blood. Therefore, the list of associations we report here is not comprehensive, and studies using other technologies and other biological sample types may reveal further associations. The specific disease areas, physiological processes, and classes of the proteins targeted by the SOMAscan assay have been described in our previous study[65]. Third, aptamer-based proteomics methods are sensitive to potential probe cross-reactivity and non-

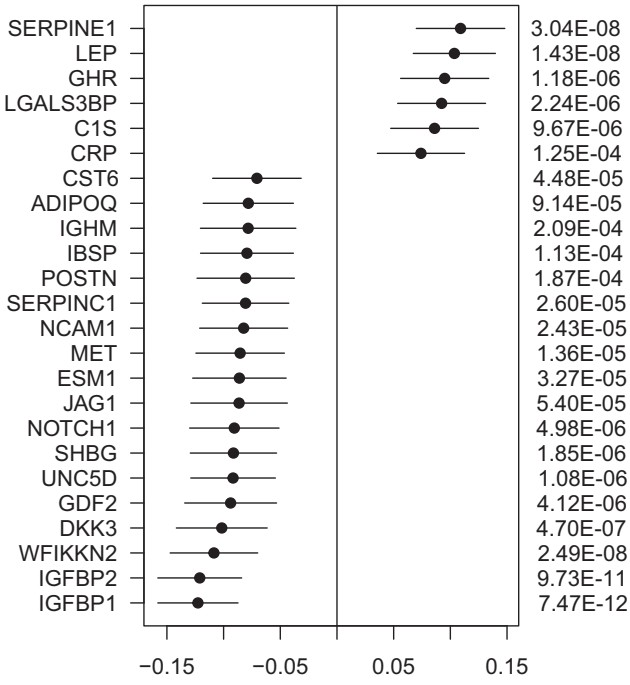

## Proteins that are an outcome of BMI (1SMR)

| Protein | p-value |
|---|---|
| SERPINE1 | 3.04E-08 |
| LEP | 1.43E-08 |
| GHR | 1.18E-06 |
| LGALS3BP | 2.24E-06 |
| C1S | 9.67E-06 |
| CRP | 1.25E-04 |
| CST6 | 4.48E-05 |
| ADIPOQ | 9.14E-05 |
| IGHM | 2.09E-04 |
| IBSP | 1.13E-04 |
| POSTN | 1.87E-04 |
| SERPINC1 | 2.60E-05 |
| NCAM1 | 2.43E-05 |
| MET | 1.36E-05 |
| ESM1 | 3.27E-05 |
| JAG1 | 5.40E-05 |
| NOTCH1 | 4.98E-06 |
| SHBG | 1.85E-06 |
| UNC5D | 1.08E-06 |
| GDF2 | 4.12E-06 |
| DKK3 | 4.70E-07 |
| WFIKKN2 | 2.49E-08 |
| IGFBP2 | 9.73E-11 |
| IGFBP1 | 7.47E-12 |

Beta from 2SLS (95% confidence interval)

**Fig. 5 Forest plot of the causal estimate of BMI on various proteins in the one-sample MR analysis (KORA).** BMI is suggested to have a causal effect on 24 out of 152 replicated proteins, using the 2SLS method. The BMI polygenic score (GPS$_{BMI}$) was used as an instrument for BMI in this analysis.

specific binding. We include the validation information for all proteins extracted from two studies in Supplementary Data 2. A full review of the limitations of SOMAscan technology such as possible epitope effects (influence of genetic variance), unspecific binding, cross-reactivity, interference with DNA-binding proteins, limited coverage of isoforms, and protein post-translational modifications have been described elsewhere[66].

As with all MR studies, limitations are statistical power, potential reverse causation, population stratification, confounding, and pleiotropy[67]. Although we took precautions to apply only valid MR instruments and report associations at conservative levels of Bonferroni significance, inference of causality should still be interpreted with caution since the validity of MR analyses is based on assumptions and has several limitations as outlined in recent reviews[67–69].

GPS capture genetic susceptibility by aggregating effects of genome-wide variation with individually modest effects. The cumulative genetic effects captured in these scores influence the plasma proteome and the risk of developing obesity. We investigated GPS$_{BMI}$ to protein associations to determine which protein levels associated with a genetic predisposition to obesity. Then, by modeling polygenic scores as proxies for obesity, we identified putatively causal effects of BMI on 24 plasma proteins. In the reverse direction, we identified six plasma proteins that have a causal effect on BMI. Complementing our findings with observations in animal models, our data suggest that LEP/LEPR, WFIKKN2, and IGFBP1 are both, a readout and driving factor for obesity, while AGER, DPT, and CTSA have a predominantly causal effect on BMI. Thus, our computational approaches combined with the assimilated experimental data, coherently suggest that the revealed associations can be bidirectional rather than strictly unidirectional. By overlaying our causal proteins with both human and mouse gene expression information and experimental evidence, we highlight potential drug targets and pathways for follow-up studies.

## Methods

**Ethics approval and consent to participate.** The project agreement for this study was granted under K060/18 g. All KORA participants have given written informed consent and the study was approved by the Ethics Committee of the Bavarian Medical Association. The QMDiab study was approved by the Institutional Review Boards of HMC and WCM-Q under research protocol number 11131/11. All study participants provided written informed consent.

**Study population (KORA).** The KORA F4 study is a population-based cohort of 3080 subjects living in southern Germany. Study participants were recruited between 2006 and 2008 comprising individuals with ages ranging from 32 to 81. Other covariates that were considered included binary diabetes information (case/control based on self-reporting or medication usage), physical activity, alcohol consumption, and smoking. For this study, aptamer-based proteomics was done using the SOMAscan platform, and the protein levels of 996 individuals, with ages ranging from 43 to 79 and consisted of 48% males, have been measured and has been described in detail elsewhere[13].

**Proteomics (KORA).** The SOMAscan platform was used to quantify the protein levels of 996 KORA individuals. Details of the SOMAscan platform have been described elsewhere[70–75]. Briefly, undepleted EDTA-plasma is diluted into three dilution bins (0.05, 1, and 40%) and incubated with bin-specific collections of bead-coupled SOMAmers in a 96-well plate format. Subsequent to washing steps, bead-bound proteins are biotinylated and complexes comprising biotinylated target proteins and fluorescence-labeled SOMAmers are photo cleaved off the bead support and pooled. Following recapture on streptavidin beads and further washing steps, SOMAmers are eluted and quantified as a proxy to protein concentration by hybridization to custom arrays of SOMAmer-complementary oligonucleotides. Based on standard samples included on each plate, the resulting raw intensities are processed using a data analysis work flow including hybridization normalization, median signal normalization, and signal calibration to control for interplate differences. One-thousand blood samples from the KORA F4 study were sent to SomaLogic Inc. (Boulder Colorado, USA) for analysis. Of the original 1000 samples, three did not have BMI information and one sample failed SOMAscan QC, leaving a total of 996 samples. Data for 1129 SOMAmer probes (SOMAscan assay V3.2) was obtained for these samples. Twenty-nine of the probes failed SOMAscan QC, leaving a total of 1100 probes for analysis (Supplementary Data 16).

**Genotyping (KORA).** The Affymetrix Axiom Array was used to genotype 3788 samples of the KORA S4 of which 996 were used in this study. After thorough quality control (total genotyping rate in the remaining SNPs was 99.8%) and filtering for minor allele frequency (MAF) > 1%, a total of 509,946 autosomal SNPs was used for imputation. Shapeit v2 was used to infer haplotypes from the available SNPs using the 1000G phase 3 haplotype build 37 genetic maps. Impute2 v2.3.2 was used for imputation. Variants with certainty < 0.95, information metric < 0.7, missing genotype data (geno 0.2), Hardy–Weinberg equilibrium (hbe) exact test $p$ value $< 1 \times 10^{-6}$, or with MAF < 0.01 were all excluded. A total of 8,263,604 variants with a total genotyping rate of 0.97 were kept for the analysis.

**Study population (QMDiab).** The Qatar Metabolomics Study on Diabetes (QMDiab) is a cross-sectional case-control study that was carried out in 2012 at the Dermatology Department in Hamad Medical Corporation (HMC Doha, Qatar). This cohort comprises 388 study participants from Arab and Asian ethnicities of which around 50% have type 2 diabetes[15]. The majority of participants were Arabs, Indians, or Filipinos. The participants were individually recruited (unrelated individuals). A subset of 356 samples having proteomics data was used in this study. The participant age ranged from 17 to 81 and included approximately 50% males.

**Proteomics (QMDiab).** The SOMAscan platform of the WCM-Q proteomics core was used to quantify a total of 1129 protein measurements in 356 plasma samples from QMDiab[13]. Protocols and instrumentation were provided and certified using reference samples by SomaLogic Inc. No sample data or probes were excluded.

**Genotyping (QMDiab).** Genotyping was carried out using the Infinium Human Omni 2.5-8 V1.2 Beadchip array for 353 samples and was previously described elsewhere[13]. After stringent quality control, 1,221,345 variants were used to impute a total of 18,829,416 variants that were used in this study. The same imputation quality metrics were used in QMDiab as in KORA.

**Polygenic score calculation.** Polygenic scores represent the quantification of an individual's inherited risk by combining the impact of thousands of common variants. Derivation, validation, and testing of the score is described

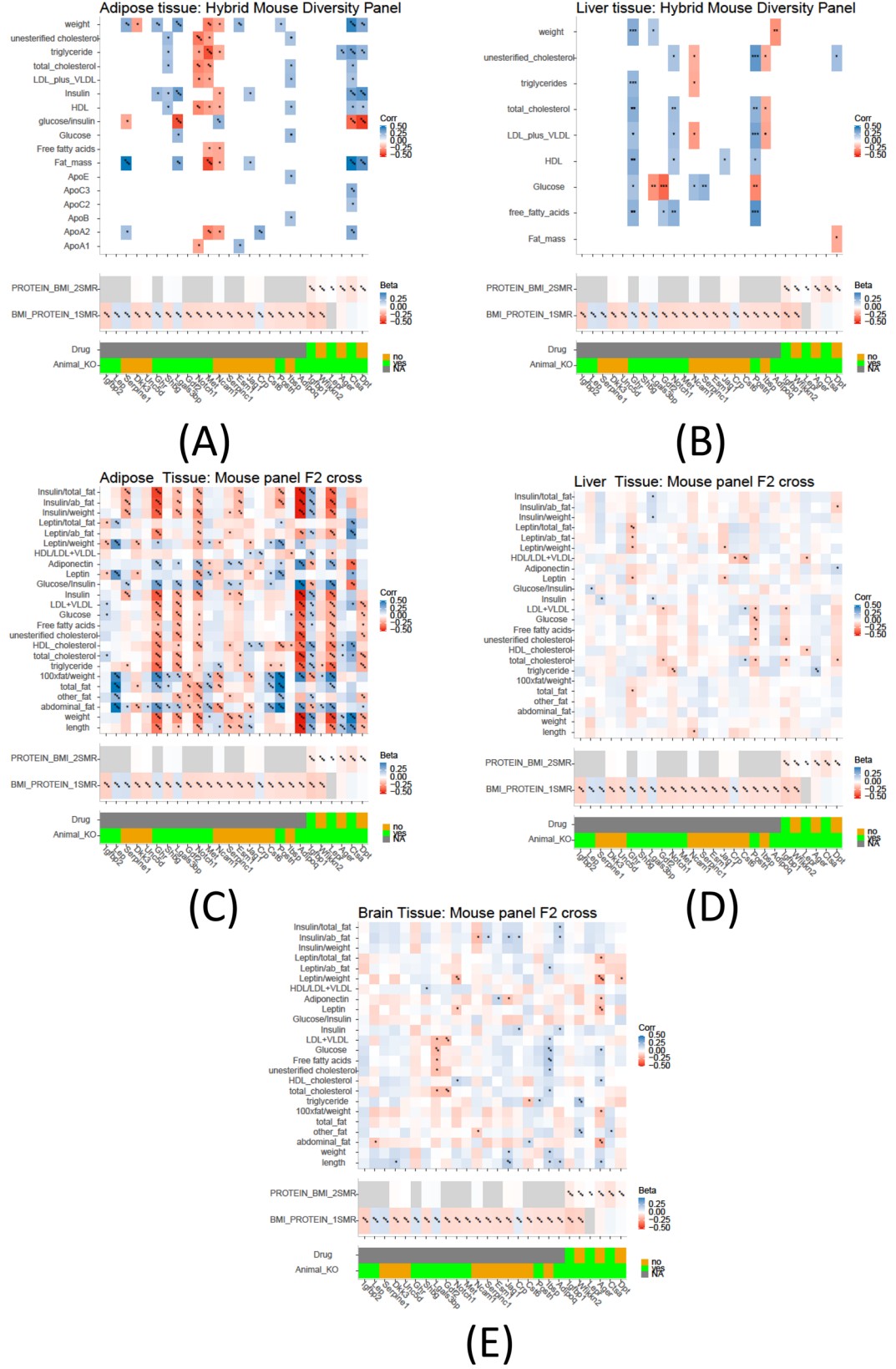

**Fig. 6 Adipose, liver, and brain tissue gene expression associations with obesity traits in mouse panels.** The bi-weight mid-correlation coefficients (median-based measures of similarity) and *p* values are shown for obesity-related traits with adipose/liver tissue gene expression levels using a threshold of *p* < 0.05 and absolute correlation coefficient >0.1 in two datasets: (A/B) the HMDP dataset consisting of 706 mice fed a standard chow diet and (C/D/E) the F2 dataset which is a cross of the inbred ApoE−/− C57BL/6J and C3H/HeJ strains fed a high fat + cholesterol diet. The significance of the correlations is as indicated (*** for *p* < 0.001, ** for *p* < 0.01, and * for *p* < 0.05). The bottom part of each plot includes the bi-directional MR results (direction and significance), whether there are existing drugs that target the tested proteins, and the animal knockout model information. Gray boxes indicate missing data.

elsewhere[17,2,76]. Briefly, the score was derived using summary statistics from a recent GWAS study for BMI covering up to 339,224 individuals and a reference panel of 503 European samples from 1000 Genomes phase 3 version 5 Ambiguous SNPs (A/T or C/G) were not included in the score derivation. A set of candidate scores were derived using the LDPred algorithm[77] which is a Bayesian method and pruning and threshold derivation strategies. Another approach that involved pruning and thresholding was used to derive additional candidate scores using an LD-driven clumping procedure in PLINK version 1.90b (–clump). These scores were then validated in another dataset. The scores were generated by multiplying the dosage of each risk allele for each variant by its respective weight, and summing across all variants in the score while incorporating genotype dosages for the uncertainty in genotype imputation. Finally, the optimal score having the best discriminative capacity based on the highest AUC with BMI as the outcome in the UK Biobank validation dataset was selected.

The derived weights of the optimal candidate BMI score were used to generate BMI scores for the 996 samples from KORA and the 353 samples from QMDiab. Scoring was carried out using PLINK version 2.0[78]. The list of variants comprising the polygenic score for BMI from Khera et al. includes 2,100,302 variants[17]. Imputed genotyping data was used, and in total, 1,583,718 (74.5%) and 1,636,172 variants (77.9%) passed QC for the GPS computation, in KORA and QMDiab respectively. The common set of variants between the two cohorts consisted of 1,565,281 variants (74.5%), and this set included 98.8% of the variants used for the score computation in KORA and 95.7% of the variants used for the score computation in QMDiab. We found the correlation between the score computed using all available variants in KORA and the intersection set to be $R^2 = 0.99$. In addition, the score computed using all available variants in QMDiab and the intersection set was $R^2 = 0.97$. Limiting the score computation to the same set of loci common among the two studies yielded minimal differences in the scores. Therefore, all available variants were included in the score computation for the two studies. Finally, the GPS$_{BMI}$ values were scaled to have a mean of 0 and a standard deviation of 1.

**Statistical analysis**. The protein measures were log2 transformed and standardized (mean = 0, sd = 1) in both KORA and QMDiab. For the BMI-protein associations in KORA, linear models were used while adjusting for age and sex. Another linear model that adjusts for age, sex, smoking, alcohol, physical activity, and diabetes was also evaluated in KORA as a sensitivity analysis. For replication of the BMI–protein associations in QMDiab, the analysis was performed using a slightly different model: (age + sex + study-specific covariates). The study-specific covariates in QMDiab consisted of diabetes status, the first three principal components (PCs) of the genotyping data (genoPC1, genoPC2, and genoPC3) along with the first three PCs of the proteomics data (somaPC1, somaPC2, and somaPC3) which were added as covariates in the analysis. Diabetes was used as a covariate in QMDiab to eliminate associations confounded by the diabetes-BMI relationship. These PCs were considered as standard covariates of the QMDiab study[15]. The genetic PCs accounted for the ethnic variability of the QMDiab cohort and the proteomics PCs accounted for a moderate level of observed cell lysis. Together, the first three genetic PCs from QMDiab explained the majority of the genetic variance (13.1%, 5.9%, and 4.0%, respectively) and effectively separated the three main ethnic groups composing QMDiab. In KORA, possible effects from population stratification have already been excluded in the previous studies[79]. Therefore, no adjustment for population structure was performed in KORA. Finally, the association of proteins with the BMI scores was carried out also using linear regression while adjusting for age and sex in KORA, and adjusting for age, sex, genetic PCs, and soma PCs in QMDiab.

To consider a BMI–protein association as replicated, we required $p < 2.72 \times 10^{-4}$ (0.05/184). We also estimated the statistical power for the replication by sampling. This was carried out for each association by randomly selecting 356 individuals from the KORA cohort (without replacement) and computing the $p$ value of association for that subset of samples. This was repeated 1000 times and the 50th smallest $p$ value from this distribution was considered to be obtained at 95% power (p95).

**Cross-reactivity of aptamers**. To ensure no BMI–protein associations were affected by binding issues, we checked all proteins for cross-reactivity (Supplementary Data 2). A list of cross-reactive proteins was obtained from the study of Sun et al.[16], who tested a subset of the SomaLogic aptamers (SOMAmers) for cross-reactivity with homologous proteins that have at least 40% sequence similarity. In addition, we assessed the specificity of the SOMAlogic assay for the proteins in our protein associations using data provided by Emilsson et al.[19], where the direct assessment of aptamer specificity using data-dependent analysis and multiple reaction monitoring mass spectrometry after SOMAmer enrichment in biological matrices.

**Mendelian randomization analysis**. We performed causal inference using both the 1SMR and 2SMR methods. All MR analyses were conducted using the MendelianRandomization library (v0.4.1)[80] and the TwoSampleMR library (v0.4.22)[81].

We used KORA data to evaluate the effect of BMI on proteins in the 1SMR, by modeling the GPS$_{BMI}$ as an instrument for BMI. After adjusting for age and sex, we computed the causal estimate of BMI on the 152 replicated proteins using the 2SLS method from the ivpack R library v.1.2.

To assess the effect of BMI on protein levels, in a 2SMR setting, we identified BMI instruments in the GIANT-UK Biobank meta-analysis[3] and extracted the corresponding SNP-exposure estimates from the INTERVAL pGWAS[16]. We selected instruments for BMI to have genome-wide significance ($p < 1 \times 10^{-8}$ and $f$-statistic > 10) and an LD clumping threshold of 0.001. We further eliminated SNPs with potential confounders using data from the UK Biobank GWAS[82]. We downloaded all variant association data with potential confounders (education, smoking, alcohol use, physical activity, etc.) from the UKBiobank GWAS (determined by the genome-wide significance of $p < 1 \times 10^{-8}$). After these filtration steps and elimination of potentially confounded SNPs, 454 BMI instruments were used. The exposure and outcome data were harmonized before performing the MR analysis by aligning the SNPs on the same effect allele for the exposure (BMI) and outcome (proteins). The 2SMR was feasible for 103 of the 152 replicated proteins for which GWAS summary statistics were available from INTERVAL. We used the IVW method to estimate the causal effect of BMI on proteins. We downloaded full protein GWAS summary statistics from INTERVAL and extracted the genetic instrument SNPs as outcome associations from this data. The 2SMR results using the IVW method suggested that BMI was causal for 9 of the 103 tested proteins, after multiple-testing correction. The causal estimates were directionally concordant with the 1SMR estimates for all significant proteins. These results were also robust to sensitivity analysis and evidence of heterogeneity or horizontal pleiotropy, based on the MR Egger analysis, was weak (Supplementary Data 11). For all tested proteins, the heterogeneity measures represented by Cochran's $Q$ statistic were not significant (p_Het > 0.05), suggesting there was no nondirectional pleiotropy. In addition, we did not find any evidence of directional pleiotropy, according to the MR–Egger intercepts test (p_Pleio > 0.05).

For the 1SMR in the protein-to-BMI direction, we identified protein instruments in KORA and also computed the corresponding SNP-outcome exposure estimates for the 152 proteins. For each protein, instruments were tested for association with protein levels in a linear regression model adjusted for age and sex. Genetic instruments were selected if their association was genome-wide significant ($f$-statistic > 10). The genome-wide instruments were filtered to include only independent signals ($r^2 = 0.001$). Out of the 152 plasma proteins, we identified one or two suitable genetic instruments for 63 proteins. After accounting for multiple testing, the 1SMR did not provide any evidence of plasma proteins having a potentially causal effect on BMI, due to lack of strong/suitable instruments (Supplementary Data 12).

Finally, for the 2SMR in the protein-to-BMI direction, the Proteome PheWAS browser (http://www.epigraphdb.org/pqtl/ accessed on April 2020)[18] was used to check if any of the proteins with suitable instruments were causal for BMI. Instrument reliability was based on pleiotropy, consistency, and colocalization scores, as defined by the authors of that study. With a single genetic variant, the estimate of the IVW reduces to the ratio of coefficients betaY/betaX or the Wald ratio.

**Gene expression analysis**. To investigate the tissue-specific role of the GPS$_{BMI}$ associated protein-coding genes with obesity, we used RNA-seq tissue expression data from both humans and mice—the Genotype-Tissue Expression (GTEx) database[83] and the Gene eXpression Database (GXD)[84] (C57BL/6J strain), respectively. The data presented and described in this paper were generated through a multi-gene query on the GTEx portal on 03/29/2020 from https://www.gtexportal.org/home/multiGeneQueryPage. Mice expression data was visualized on the GXD portal where expression data were processed using the Morpheus heat map and visualization and analysis tool created by the Broad Institute from http://www.informatics.jax.org/expression.shtml.

**Identifying the tissue-specific role of GPS-associated proteins in obesity using mouse databases**. Mouse orthologs were identified for the genes encoding the causal/consequential proteins using the MGI database (http://www.informatics.jax.org/)[26]. An ortholog was present for 26 proteins. Two reference mouse databases were used to identify correlations between adipose, liver, and brain tissue expression of these proteins and the relevant obesity traits in mice: (a) the HDMP ($n = 706$ mice from 100 well-characterized inbred strains fed with a standard chow diet)[24], and (b) an F2 cross of the inbred ApoE−/− C57BL/6J and C3H/HeJ strains ($n = 334$ mice that were fed with a high fat and cholesterol diet from 8 to 16 weeks of age and sacrificed at 24 weeks of age)[25]. Publically available data from the systems genetic resource was downloaded and used to search for gene-trait correlations in adipose and liver tissues from https://systems.genetics.ucla.edu/[24]. The adipose, liver, and brain tissue expression data of the 334 F2 cross mice was accessed using the publically available dataset Sage BioNetworks at https://www.synapse.org/#!Synapse:syn4497[25]. The weight mild correlation coefficients (bicor) is a similarity measure between samples based on the median which is less sensitive to outliers and provides a robust alternative to similarity metrics like Pearson correlation. The bicor coefficients and the $p$ values for the association of gene expression levels and the selected obesity relevant traits were computed using the WGCNA R package[85]. Gene to trait correlations was filtered to only include absolute correlation coefficients > 0.1 and $p$ value < 0.05 for both datasets.

**Reporting summary**. Further information on research design is available in the Nature Research Reporting Summary linked to this article.

## Data availability

All summary statistics and association data for KORA and QMDiab are available in Supplementary Data 2, 3, 5, and 8. The informed consent given by the study participants does not cover the posting of participant-level phenotype and genotype data in public databases. However, data are available upon request from KORA-gen (http://epi.helmholtz-muenchen.de/kora-gen). Requests for both KORA and QMDiab are submitted online and are subject to approval by the KORA board. Publically available datasets from the following databases are available at these web links:

DrugBank database: https://www.drugbank.ca
Proteome PheWAS browser: http://www.epigraphdb.org/pqtl
Phenoscanner: http://www.phenoscanner.medschl.cam.ac.uk
Genotype-Tissue Exxpression (GTEx): https://www.gtexportal.org/home/multiGeneQueryPage
Gene eXpression Database (GXD): http://www.informatics.jax.org/expression.shtml
Mouse Genome Informatics (MGI) database: http://www.informatics.jax.org/
Sage BioNetworks dataset: https://www.synapse.org/#!Synapse:syn4497

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

## Acknowledgements

This work was supported by the Biomedical Research Program at Weill Cornell Medicine in Qatar, a program funded by the Qatar Foundation. K.S. is also supported by QNRF grant NPRP11C-0115-180010. The KORA study was initiated and financed by the Helmholtz Zentrum München—German Research Center for Environmental Health, which is funded by the German Federal Ministry of Education and Research (BMBF) and by the State of Bavaria. Furthermore, KORA research was supported within the Munich Center of Health Sciences (MC-Health), Ludwig-Maximilians-Universität, as part of LMUinnovativ. The statements made herein are solely the responsibility of the authors.

The KORA-Study Group consists of A. Peters (speaker), J. Heinrich, R. Holle, R. Leidl, C. Meisinger, K. Strauch, and their co-workers, who are responsible for the design and conduct of the KORA studies. We gratefully acknowledge the contribution of all members of field staff conducting the KORA F4 study. We thank the staff of the HMC dermatology department and of WCM-Q for their contribution to QMDiab. The Genotype-Tissue Expression (GTEx) project was supported by the Common Fund of the Office of the Director of the National Institutes of Health, and by NCI, NHGRI, NHLBI, NIDA, NIMH, and NINDS. Finally, we are grateful to all study participants of KORA and QMDiab for their invaluable contributions to this study.

## Author contributions

Conceived and designed the study: S.B.Z., S.S., H.G., and K.S. Performed experiments: S.B.Z. and S.S. Analyzed data: S.B.Z., S.S., H.G., and K.S. Contributed reagents/materials/analysis tools: M.M., P.R.M., M.A.E., W.R., J.G., K.S., A.P., C.G., and M.W. Wrote the paper: S.B.Z., S.S., H.G., and K.S. All authors discussed the results and reviewed the final paper.

## Competing interests

The authors declare no competing interests.
