## [Peer Review File · Nature Communications]

Reviewer #1 (Remarks to the Author):

Zaghlool and colleagues examined associations and causal directions of effect between body mass index (BMI) and levels of 921 blood proteins. They initially identified association in almost 1000 individuals from the KORA F4 study and reported the 152 that replicated in another study. Polygenic risk scores for BMI were associated with 19 proteins, 5 of which replicated in a smaller study. Mendelian randomization tests suggested that 3 proteins affect BMI levels, that BMI affects 21 protein levels, and that 3 proteins both affect BMI and are affected by BMI. Based on bioinformatics and literature review of these 27 proteins, several are known to be involved in obesity pathways and others might offer new mechanistic insights.

The analysis of proteomic data is relatively new. The study design using initial replication of the association analyses is rigorous, and the strategies to consider the direction of effect are an especially valuable contribution to the field. For example, the explanations about the effects and limitations of sample size on precision and Mendelian Randomization are important to help interpret the causality inferences.

In terms of biology, most of the associations presented confirm previous reports, as cited, including the associations with LEP, the 'tail effect' for polygenic risk score association with BMI, and several genes reported to show causal effects on BMI.

Major comments:

1. For variants with very large effects on BMI, especially lower frequency variants, a GWAS signal may extend more than 1 Mb. For example, the analysis described on lines 154-156 may not remove the cis effects around the LEP gene encoding leptin. Do the association between GPS and BMI and the extreme tail effect persist if a larger region around LEP is excluded from the GPS?
2. Figures 3 and 4 focuses on LEP, which is secreted from adipocytes and has a well-established relationship with BMI. The manuscript could be improved by highlighting less well-established BMI-protein associations.
3. The last paragraph of results (lines 246-255) appears to suggest that the six identified genes would be good drug targets to treat obesity, yet the last drug seems to increase growth, which would exacerbate obesity. Given the focus on direction of effect in the article, the goals and results in this paragraph should be clarified.

Minor comments:

1. Association analyses with QMDiab included principal components to account for variation in genotypes and proteomics technical factors, but analyses in KORA did not. What analyses were done in KORA to consider these potential sources of variation?
2. Lines 121, 124, and 150: when results are reported as significant or replicated, please clarify whether the direction of effect is required to be consistent with the initial association.
3. In Figure 1, why are 28 proteins examined in the last step if the previous step identifies $24 + 6 - 3$ overlap = 27 proteins? Similarly, why does the text line 214 refer to 28 proteins and not 27?
4. Consider including the sample sizes and p-value thresholds in Figure 1.
5. The bottom sections of Figure 6 (SMR, Drug, Animal KO) are not described in the legend. This figure perhaps should be moved to the supplement.
6. Text lines 217 - 221: how many proteins are considered part of each cluster?

Reviewer #2 (Remarks to the Author):

In this paper Zaghlool et al. present data that tries to disentangle some of the causal molecular mechanisms that contribute to obesity. They use proteins as intermediate phenotypes to capture differences between obese and non-obese subjects by performing a proteome wide association study in a German cohort of approximately 1000 subjects. They then use polygenic risk scores to capture the genetic contribution to weight (BMI). Lastly, they use a Mendelian Randomization process to investigate the causal driver of the observed associations i.e. does BMI drive the observed differences in protein levels or do the protein levels drive the differences in BMI.

General comments:

Overall the methodology used for the different analyses (protein-BMI association, polygenic risk score estimation and Mendelian Randomization) is sound and state of the art for these kinds of studies. The approach to use a polygenic risk score to capture all the genetic information pertaining to BMI, rather than single SNP analyses increases the power. The elucidation of the biology leading to obesity is a major challenge to advance the field and using intermediate phenotypes like proteins, metabolites and transcriptomics is a powerful means that can provide new mechanistic insights.

However, in its current form the article has a number of weaknesses to really provide causal understanding of the molecular mechanisms involved in the development of obesity.

Minor comments:

There are a number of inconsistencies in the use of the terms BMI and obesity in the introduction in describing the genetic landscape of BMI. In several parts of the introduction (e.g. line 74/75) the two terms are used as equivalent, which they're not. The GWAS studies mentioned are all QTL studies with BMI as the outcome variable. Obviously BMI and obesity are strongly correlated (and BMI is still used to define obesity clinical status) they're not the same and should not be used thus. In parts the manuscript is difficult to read, especially in the results section. There seems to be some inconsistency between the number of proteins investigated in the Mendelian randomization analysis (24 proteins in a BMI-protein direction and 6 in protein-BMI direction) but then in the next paragraph only 28 proteins are investigated? Also, although all results are clearly identified in the tables and figures, there is basically no description of the main findings in the text itself making it necessary to jump back and forth between the tables/figures (often supplementary) to put the results into context. It would be preferable to reiterate the main results findings also in the text.

Specific comments:

Cohorts. The authors used data from three different cohorts. The Kora study was used as the initial exploratory data set and the QMdiab and data from the publicly available Interval study for replication. The use of the large Interval study provides a great source for the replication of results from the Kora study.

However, there are several issues with using the QMdiab study. Firstly this is a much smaller study (only about 360 subjects) and is potentially confounded by the presence of type 2 diabetic subjects. The authors state that they tested potential confounding effects and didn't find any, which seems strange given the strong correlation between T2D and obesity. This might reflect a lack of power in the analysis. The potential lack of power is also cited by the authors for several of the analyses. Secondly, this is a highly admixed population of Middle Eastern and Asian origin (see also comments about genetic risk score estimates). In the end only 20% of the protein associations can be replicated in QMdiab. In this reviewer's opinion, the QMdiab cohort does not contribute significantly to the analyses and conclusions of the study and could be removed.

Genetic risk score. The authors used data from the UK biobank to estimate the weights applied to building the PRS. Whereas it can be argued that the German Kora cohort is reasonably genetically similar to use these weights, this is not the case for the QMdiab study, which is a highly admixed population of Middle Eastern and Asian origin. There is no a priori reason that the weight estimates should apply in this population. An unweighted score (based on simple allele count) could also be computed and results compared. Also, for the analyses the authors include only the first three genetic PCs. This is less than what is habitually used. The authors should provide a rationale for choosing the first 3 PCs (by providing variance explained for instance) and a plot for the first 2 PCs with different colors for each cohort. This may also be of specific importance for the Middle East population included in this study as some Middle Eastern populations show a high degree of co-sanguinity (up to 30%) that strongly influence the genetic substructure of the population. This is also important in terms of the Mendelian randomization strategy that assumes random mating. It would also be of interest to see the impact of SNPs used in the PC and the genetic score. How many of them are in both components? Removal of genetic score SNPs from the PCs as well as SNPs in LD might improve the association results.

Genotyping. The Kora and QMdiab cohorts were genotyped on different chips (Affymetrix and Illumina respectively). It seems that the authors used different imputed SNP sets for the PRS

calculations. Did they create a combined minimal set of imputed SNPs for both cohorts? In the absence of information of the LD structure it is difficult to interpret the GPS-BMI results (do they pertain to the same set of loci or are they different)?

Functional interpretation of GPS effects. The authors perform analyses to dissect the potential SNP effects (cis- or trans) for the variants in the genetic score and they use a distance metric to remove potential strong cis-effects to evaluate if the overall effects are mostly cis- or trans-driven. They should provide a short summary of the number of potential cis- vs trans effects. Also, it has been shown that cis- SNPs can have effects over large distances due to the 3-dimensional structure of DNA. The metric used (a fixed 1 Mbase interval) is a very crude solution to remove potential cis-acting SNPs. A LD based approach and/or an approach using publicly available eQTL data for the genetic regions in question would be more appropriate. It seems also strange that the cis- SNPs seem not to have a significant effect on the associations, which would mean that most cis-effects are weak and/or that there were not many cis-effect SNPs in the GPS, both of which would be contrary to the existing protein QTL literature (e.g. Chick et al. 2016) and should be discussed. If there are such strong trans- effects it would be worthwhile to investigate and mention and discuss some of the associated gene loci, as they may hold the actual mechanistic clues.

Tail effect. The authors report a significant tail effect for the distribution of the GPS-BMI score variants and a similar effect for the 19 GPS associated proteins. This is an interesting observation and it could mean that there is a threshold that distinguished "mainly" genetic obesity from "more" environmental obesity. This might also explain some clinical observations that people with higher BMI's seem to have an earlier onset of obesity than those with overweight/moderate obesity. This might be important for the treatment strategies of these individuals. Do the authors observe a similar tail effect if they use all 152 BMI associated proteins? If not, this would indeed argue for a very strong genetic effect on the extremes of the BMI distribution.

Proof of causality. Out of 152 BMI associated proteins 30 seem to be causally linked to BMI (24 driven by BMI, 6 driving BMI). Is there any functional connection between these 30 proteins with the 120 proteins for which no effects were observed? In this context also the limitations of the MR in determining causality should be discussed in more detail.

The term causality as it is used by the authors needs some clarification and should be discussed. For 24 proteins BMI seems to be the driver for the observed increase/decrease of protein concentrations. But BMI itself is a complex, heterogeneous phenotype. One example is leptin. Leptin increases with BMI as a result of the expanding fat tissue combined with a peripheral resistance to leptin signaling. BMI and fat mass are obviously associated, explaining the strong correlation with leptin. However, this does not explain the causal molecular mechanism that leads to leptin resistance and the inhibition of the feedback loop that should result in increased satiety signals as fat mass expands. Thus stating that BMI causes high leptin levels does not provide mechanistic insights.

Discussion. Generally the discussion lacks in a critical assessment of the potential limitations of the analyses performed and should be reworked in this respect. Although not a major point, the limitations of the Somalogic technology should also be discussed (e.g. the fact that this technology provides relative abundance but not absolute concentrations, potential bias because of the choice of proteins etc.).

The focus of the leptin and leptin receptor animal models in the discussion seems strange, given that the authors have found some potentially interesting new results for some of the proteins for which there are mouse ko. Indeed, some of the text in the results section is more descriptive of these animal findings and should be moved to the discussion and be extended. The discussion about the potential drug targets is highly speculative as there is no evidence that these targets can be used in obesity treatment from the available data and should be either shortened or removed.

Reviewer #3 (Remarks to the Author):

The paper presents a series of analyses from three cohorts to examine the relationship between plasma protein levels and obesity, as measured by body mass index (BMI). The work first

establishes association between protein levels and BMI, then tested these proteins for association with a genome-wide polygenic risk score for BMI developed from published data. This was then followed by bi-directional MR analysis to infer causal relationships between protein levels and obesity. Finally, through bioinformatics and literature search, the authors posit potential functional roles for implicated/causal proteins.

The paper asks a series of clear and understandable questions about circulating protein levels and their impact on/from obesity. While the sets of analyses make good sense and address appropriate questions for these data, several aspects need to be clarified before publication.

Comments:

RE: Relation of associated proteins to GWAS/eQTL loci.

- 1) How many proteins fall in known BMI loci or have eQTLs near BMI loci
- 2) Could be/have been considered best candidate in the regions (closest gene, etc.)
- 3) Does this data help inform on "causal" loci in GWAS regions? Provide new insights?

RE: Protein analysis

- 1) Are there specific kinds of proteins most/least amenable to analysis with this method? How would this impact the ability to discover associations, particularly for BMI.
- 2) Relevance/generalizability of plasma protein levels for obesity generally and vis a vis other obesity-related tissues.

RE: MR analyses

- 1) MR assumes that genetic variants have no other effect on the outcome of interest (either BMI or proteins) except through the outcome risk factor. As described, I'm not convinced that either the BMI GPS or the individual protein GPS's satisfy this assumption. If they do, please clarify in the methods how the instruments were derived to stasify any effects not through the risk factor. If they do not satisfy this assumption, analysis should be redone to account for this assumption.
- 2) MR also assumes there are no confounders of the genetic variant-outcome association, but no attempt is made to adequately assure that this assumption is also satisfied.

RE: Figure 3

- 1) The X axis reflects deciles (0-10, 11-20, etc.), not percentiles, correct?
- b) It seems odd to see essentially no effect on LEP levels outside of top and bottom deciles.
- c) Does the GPS include SNPs associated with LEP levels?

Given the comments on environmental contributions to BMI, it might be useful to comment on the lack of effect of "environmental" risk factors on the protein/BMI associations?

Similarly, there is no discussion of the relevance of the broadly expressed versus sporadically expressed tissues and what importance to place on these results. Could pathway analyses be useful?

Given a known and strong role for the CNS in BMI/obesity biology, why were comparisons only done in adipose and liver tissue for follow-up studies. Why not consider roles for CNS tissues too?

The authors offer competing explanations for diminished performance in the QMdiab study, poor performance of European GPS and power. Both are plausible, but no effort is made to distinguish between the two possibilities? Can this be determined empirically?

Point-by-point response to the referees' comments

"Elucidating the genetic risk of obesity through the human blood plasma proteome"

Zaghlool et al.

Reviewer #1 (Remarks to the Author):

Zaghlool and colleagues examined associations and causal directions of effect between body mass index (BMI) and levels of 921 blood proteins. They initially identified association in almost 1000 individuals from the KORA F4 study and reported the 152 that replicated in another study. Polygenic risk scores for BMI were associated with 19 proteins, 5 of which replicated in a smaller study. Mendelian randomization tests suggested that 3 proteins affect BMI levels, that BMI affects 21 protein levels, and that 3 proteins both affect BMI and are affected by BMI. Based on bioinformatics and literature review of these 27 proteins, several are known to be involved in obesity pathways and others might offer new mechanistic insights.

The analysis of proteomic data is relatively new. The study design using initial replication of the association analyses is rigorous, and the strategies to consider the direction of effect are an especially valuable contribution to the field. For example, the explanations about the effects and limitations of sample size on precision and Mendelian Randomization are important to help interpret the causality inferences.

In terms of biology, most of the associations presented confirm previous reports, as cited, including the associations with LEP, the 'tail effect' for polygenic risk score association with BMI, and several genes reported to show causal effects on BMI.

Response: We thank the reviewer for their interest in our study. We addressed the specific concerns as described below. We hope we answered all questions and provided satisfactory clarification where needed.

Major Comments

1. For variants with very large effects on BMI, especially lower frequency variants, a GWAS signal may extend more than 1 Mb. For example, the analysis described on lines 154-156 may not remove the cis effects around the LEP gene encoding leptin. Do the association between GPS and BMI and the extreme tail effect persist if a larger region around LEP is excluded from the GPS?

Response:

*To answer the reviewer's question, we increased the window size to 100 Mb to account for GWAS signals that may extend more than 1 Mb, as well as excluded any known pQTLs. We found that the tail effect still persists when this larger region around LEP is excluded from the GPS. The same was also true for all other 18 proteins that were associated with the GPS. We have updated the text to reflect the replacement of the 1Mb window with 100 Mb and present the results of this analysis in **Supplementary Table 7**.*

2. Figures 3 and 4 focuses on LEP, which is secreted from adipocytes and has a well-established relationship with BMI. The manuscript could be improved by highlighting less well-established BMI-protein associations.

Response: Following the reviewer's suggestion we now highlight other well-established BMI-protein associations by adding the plots for IGFBP1 and WFIKKN2 to both **Figures 3 and 4**. In addition, similar plots for the remaining 19 proteins, are included in **Supplementary Figure 3**.

We shortened the parts that focus on LEP and added the following on IGFBP1, IGFBP2, AGER, CTSA, DPT, and AGER to the discussion:

"A global IGFBP1 deletion in mice showed a significant increase in body weight and body fat mass [50]. Interestingly, epigenetic regulation of IGFBP2 has also been suggested to play a role in abdominal obesity [51].

WAP, Kazal, immunoglobulin, Kunitz and NTR domain-containing protein 2 (WFIKKN2) is a protease-inhibitor that contains multiple distinct protease inhibitor domains [52]. WFIKKN2 encodes GASP1 (Growth and differentiation factor-associated serum protein-1) [53] and WFIKKN2 is an inhibitory TGF- β binding protein. Animal knockout models for WFKINN2/GASP1 have also shown a significant increase in body weight, particularly in muscle mass [54].

Further, AGER knock out models showed weight gain and increased plasma cholesterol [55], and CTSA knock out mice presented with thick skin that contained enlarged hyperplastic epidermal glands as well as a reduction in dermal fat [56]. The circulating soluble receptor for advanced glycation end products (AGER) is negatively associated with BMI [57], as we also observed in KORA. In addition, recent evidence suggests a novel role of adipokine dermatopontin (DPT) in obesity by regulation of adipose tissue remodeling and inflammation [58]. A DPT knockout mouse model showed increased subcutaneous adipose tissue [59], and effects on skin elasticity, dermis thickness, and collagen accumulation.

GASP1/WFKINN2 has mainly been involved in skeletal and muscle fiber development in the heart [60]. Higher WFIKKN2 protein levels were associated with lower levels of fasting insulin, triglycerides, HOMA-IR and visceral fat [61] suggesting a protective role against metabolic dysregulation.

In addition, global overexpression of WFIKKN2/GASP1 resulted in a significant increase in body weight in mice [54]. Interestingly, our findings were consistent with a recent SOMAscan protein study of type 2 diabetes in AGES- Reykjavik, where WFIKKN2 was reported to be potentially causal for type 2 diabetes [62], independent from BMI. Furthermore, WFIKKN2 was suggested as a potentially causal candidate for type 2 diabetes, in a second study from INTERVAL that associates the diabetes risk score with proteins [63], after adjusting for age, sex and technical covariates. Lastly, genetic variants in the WFIKKN2 locus (cis-pQTLs) showed regulation of GDF8/11 at the protein level in a trans-pQTL manner [16]. Thus, the plasma levels of GDF8/11 and WFIKKN2 are strongly controlled by genetics. Genetically supported targets could be more successful than those without genetic support in clinical practice [64], suggesting that WFIKKN2 is a potential target that would modulate GDF8/11 function, as suggested in Sun et al. [16]."

3. The last paragraph of results (lines 246-255) appears to suggest that the six identified genes would be good drug targets to treat obesity, yet the last drug seems to increase growth, which would exacerbate obesity. Given the focus on direction of effect in the article, the goals and results in this paragraph should be clarified.

Response:

We agree with the reviewer that drugs that exacerbate growth are not suitable for treatment of obesity. However, the pathways targeted by such drugs are still of interest, for example, a mediator upstream of the associated protein may potentially reduce this protein's levels. To avoid any confusion, we have clarified the goal of this section by mentioning the goal of identifying drug targets in the results:

"Lastly, and in order to identify drug targets for the potential treatment of obesity, we used the DrugBank database [30] to search for existing drugs that target the six proteins that were causal for BMI."

and

"Although the latter drug actually exacerbates growth, the affected pathway(s) may still be considered a potential target for medical intervention."

Minor Comments:

1. Association analyses with QMDiab included principal components to account for variation in genotypes and proteomics technical factors, but analyses in KORA did not. What analyses were done in KORA to consider these potential sources of variation?

Response: KORA has been used in numerous previous studies, many of which investigated such potential sources of variation. It has been shown that KORA does not show any substantial degree of population stratification and PCs are generally not considered as necessary covariates in studies with the KORA population. We clarified this point by adding the following to the methods section.

"In KORA, possible effects from population stratification have already been excluded in previous studies [79]. Therefore, no adjustment for population structure was performed in KORA."

2. Lines 121, 124, and 150: when results are reported as significant or replicated, please clarify whether the direction of effect is required to be consistent with the initial association.

Response: We apologize for the confusion. When we refer to replication we always include consistency of directionality as a criterion. We clarified this as follows:

"Of the 184 proteins, 150 BMI-protein associations (81.5%) were replicated (both significant and directionally concordant) in INTERVAL, after Bonferroni correction ($p < 2.72 \times 10^{-4}$; $0.05 / 184$). In QMDiab, 37 (20.1%) of the BMI-protein associations were replicated after Bonferroni correction, while a further 131 proteins (71.2%) were directionally concordant, but did not reach the required significance level for replication. In total, 152 BMI-protein associations were replicated in at least one study – specifically, 35

associations replicated in both studies, 115 associations replicated only in INTERVAL, and two new associations replicated only in QMDiab (THBS2 and ANGPT2).”

3. In Figure 1, why are 28 proteins examined in the last step if the previous step identifies 24 + 6 - 3 overlap = 27 proteins? Similarly, why does the text line 214 refer to 28 proteins and not 27?

Response: We are sorry for the confusion. We corrected the text so that the numbers add up (24 + 6 - 2) =28:

“In summary, we found that BMI had a causal effect on the levels of 24 proteins, while six proteins had a potentially causal role in the development of obesity, two of which are suggested to have roles in both directions (IGFBP1 and WFIKKN2). LEPR/LEP also showed roles in both directions (LEPR-to-BMI, and BMI-to-LEP).”

4. Consider including the sample sizes and p-value thresholds in Figure 1.

Response: We modified **Figure 1** to include the sample sizes and p-values thresholds accordingly.

5. The bottom sections of Figure 6 (SMR, Drug, Animal KO) are not described in the legend. This figure perhaps should be moved to the supplement.

Response: We added a description for the bottom sections of **Figure 6** by adding this text:

“The bottom part of each plot includes the bi-directional MR results (direction and significance), whether there are existing drugs that target the tested proteins, and the information about animal knockout model phenotypes. Grey boxes indicate missing data.”

We prefer to keep this figure in the main paper as we feel that it conveys important integrative information about all the results of the paper (correlation of expression and obesity traits, MR, drug targets, and animal knock outs). In addition, please note that the third reviewer actually asked to add additional information to this Figure, which we did (please see our response to reviewer #3 for details).

6. Text lines 217 – 221: how many proteins are considered part of each cluster?

Response: We clarified this in the text.

“The first cluster consists of seven proteins....”

“The second cluster consisting of 13 proteins...”

Reviewer #2 (Remarks to the Author):

In this paper Zaghlool et al. present data that tries to disentangle some of the causal molecular mechanisms that contribute to obesity. They use proteins as intermediate phenotypes to capture differences between obese and non-obese subjects by performing a proteome wide association study in a German cohort of approximately 1000 subjects. They then use polygenic risk scores to capture the genetic contribution to weight (BMI). Lastly, they use a Mendelian Randomization process to investigate the causal driver of the observed associations i.e. does BMI drive the observed differences in protein levels or do the protein levels drive the differences in BMI.

General comments:

Overall the methodology used for the different analyses (protein-BMI association, polygenic risk score estimation and Mendelian Randomization) is sound and state of the art for these kinds of studies. The approach to use a polygenic risk score to capture all the genetic information pertaining to BMI, rather than single SNP analyses increases the power. The elucidation of the biology leading to obesity is a major challenge to advance the field and using intermediate phenotypes like proteins, metabolites and transcriptomics is a powerful means that can provide new mechanistic insights. However, in its current form the article has a number of weaknesses to really provide causal understanding of the molecular mechanisms involved in the development of obesity.

Response: We thank the reviewer for their interest in our study. We addressed the specific concerns as described below and hope that we answered all questions and provided satisfactory clarification where needed.

Minor Comments:

There are a number of inconsistencies in the use of the terms BMI and obesity in the introduction in describing the genetic landscape of BMI. In several parts of the introduction (e.g. line 74/75) the two terms are used as equivalent, which they're not. The GWAS studies mentioned are all QTL studies with BMI as the outcome variable. Obviously BMI and obesity are strongly correlated (and BMI is still used to define obesity clinical status) they're not the same and should not be used thus.

Response: We agree with the reviewer and now replaced the term "obesity" with "BMI" where relevant.

In parts the manuscript is difficult to read, especially in the results section. There seems to be some inconsistency between the number of proteins investigated in the Mendelian randomization analysis (24 proteins in a BMI-protein direction and 6 in protein-BMI direction) but then in the next paragraph only 28 proteins are investigated?

Response: Please refer to our response to reviewer # 1.

Also, although all results are clearly identified in the tables and figures, there is basically no description of the main findings in the text itself making it necessary to jump back and forth between the tables/figures (often supplementary) to put the results into context. It would be preferable to reiterate the main results findings also in the text.

Response: Following the reviewer's suggestion, we now describe the main findings in the text where necessary and attempt to put results into context. Please see the track-changed document for details. We ensured that every Table or Figure that is referred to in the Results section has a brief description of the findings. For example:

"The study descriptive statistics for the 996 individuals are provided in **Supplementary Table 1**. We did not observe any significant differences ($p < 0.001$) in smoking and alcohol consumption between individuals with $BMI \geq 30$ and $BMI < 30$."

"107 proteins were negatively correlated with BMI while 77 were positively correlated (Figure 2). The full summary statistics for the basic model (adjusting for age and sex only) are presented in **Supplementary Table 2**."

"We tested the influence of a number of potential confounders on the BMI-protein associations, specifically, smoking status, alcohol consumption, physical activity, and binary diabetes state (**Supplementary Table 3**). We did not observe any substantial effect of confounding by these factors on the BMI-protein association results, and all Bonferroni significant protein-BMI associations found in the full model were also significant in the model where only age and sex were included as covariates."

"We confirmed the BMI-protein associations using the published INTERVAL associations [16] and additionally attempted replication of these 184 associations in the multi-ethnic QMDiab study (**Supplementary Table 4**). Of the 184 proteins, 150 BMI-protein associations (81.5%) were replicated (both significant and directionally concordant) in INTERVAL, after Bonferroni correction ($p < 2.72 \times 10^{-4}$; $0.05 / 184$). In QMDiab, 37 (20.1%) of the BMI-protein associations were replicated after Bonferroni correction, while a further 131 proteins (71.2%) were directionally concordant, but were not sufficiently powered for replication."

"The Pearson correlation for the effect sizes is $R = 0.92$ between KORA and INTERVAL, and $R = 0.84$ between KORA and QMDiab. (**Supplementary Figure 1**)."

"The GPS_{BMI} was strongly associated with BMI in KORA ($p = 2.32 \times 10^{-43}$), and was also significant in the multi-ethnic QMDiab study ($p = 5.54 \times 10^{-4}$) (**Supplementary Figures 2a and 2b**)."

"19 proteins were associated with GPS_{BMI} in KORA after accounting for multiple testing ($p < 5.43 \times 10^{-5}$; $0.05/921$) (**Supplementary Table 5**)."

"All 19 GPS_{BMI} – associated proteins were also strongly associated with BMI in KORA (**Table 1**). The regression coefficients for the BMI-protein associations and the GPS_{BMI} -protein associations were directionally concordant. The strongest protein association with BMI (LEP; $p = 3.34 \times 10^{-136}$) was also the strongest with GPS_{BMI} ($p = 1.32 \times 10^{-12}$), followed by IGFBP1, IGFBP2, SERPINE1, and WFIKKN2."

"We replicated the analysis in QMDiab to evaluate the applicability of a polygenic score derived from European participants to a cohort of mixed non-Caucasian ethnicity. Using linear regression and adjusting for age, sex, and study-specific covariates (described in Methods), five log2 transformed proteins remained

significantly associated with GPS_{BMI} after Bonferroni correction in QMDiab ($p < 0.05/19$; 2.63×10^{-3}) (NOTCH1, C5, NCAM1, CRP, and SERPINC1), while another six proteins (LEP, IGFBP1, WFIKKN2, UNC5D, MET, RARRES2), were nominally associated with concordant directionality ($p < 0.05$) (**Supplementary Table 6**).

“All of the 19 protein associations with GPS_{BMI} remained significant after eliminating potential cis-pQTL effects (**Supplementary Table 7**).”

“The effect estimate was in fact much stronger at the extremes of the distribution (Figure 3), which agrees with previous reports of this “tail effect” [9]. A tail effect is observed when the ratio of the effect at the tails to the effect of the entire distribution is greater than 1. To evaluate this tail effect in our study, we stratified the 996 KORA study samples based on GPS_{BMI} percentiles. We found a steeper slope with respect to BMI and several protein measures (LEP, WFIKKN2, IGFBP1) at the lower and upper extremes of the distribution.”

“To investigate whether a similar tail effect can be observed for the associations between GPS_{BMI} and blood circulating proteins, we compared the different effect sizes and significance levels at various percentiles of the GPS_{BMI} distribution for all proteins (**Supplementary Table 8**), including the full dataset ($N=996$), the 25th vs. 75th percentiles, the 20th vs. 80th percentiles, the 15th vs. 85th percentiles, the 10th vs. 90th percentiles, and the 5th vs. 95th percentiles.”

“We found that the effect of GPS_{BMI} on the log2 transformed LEP, IGFBP1, and WFIKKN2 was almost quadrupled in the 5% tail of the population compared to the full data (Figure 4). Individuals in the extreme tail of the GPS_{BMI} distribution showed an over-proportionally increased genetic predisposition for developing obesity [17].”

“We found a similar tail effect for all 19 GPS_{BMI} -associated proteins (**Table 2**) (**Supplementary Figure 3**).”

“We further tested whether a similar tail effect was observed for the remaining 133 BMI-associated proteins (**Supplementary Table 9**). 81 out of 133 proteins were associated with BMI ($p < 5.43 \times 10^{-5}$), but weakly associated with the GPS_{BMI} ($0.05 \leq p \leq 6.30 \times 10^{-5}$)...”

“In all applicable cases (including nominal associations), we found consistency in the MR effect directions between the 1SMR and 2SMR, and in both directions of the MR (BMI to protein, and protein to BMI) (**Supplementary Table 10, 11, 12, 13**).”

“To assess whether proteins are causally affected by BMI in the direction (BMI-to-protein) or vice versa (protein-to-BMI), we carried out bi-directional Mendelian randomization investigations. We initially conducted both, a one-sample (1SMR) and a two-sample (2SMR) Mendelian randomization analysis, and in both directions (**Table 3**). MR analysis results are presented using the 2SLS method for the 1SMR, and using the IVW method for the 2SMR. In the BMI-to-protein direction, we used GPS_{BMI} as an instrument for BMI. Our results indicated that the 1SMR had higher statistical power than the 2SMR in identifying significant MR associations...”

“In all applicable cases (including nominal associations), we found consistency in the MR effect directions between the 1SMR and 2SMR, and in both directions of the MR (BMI-to-protein, and protein-to-BMI) (**Supplementary Table 10, 11, 12, 13**).”

“BMI was suggested to have a causal effect on 24 proteins, after correction for multiple testing ($p < 0.05/152 = 3.29 \times 10^{-4}$) (Figure 5, Supplementary Table 10).”

Specific comments:

Cohorts. The authors used data from three different cohorts. The Kora study was used as the initial exploratory data set and the QMDiab and data from the publicly available Interval study for replication. The use of the large Interval study provides a great source for the replication of results from the Kora study. However, there are several issues with using the QMDiab study. Firstly, this is a much smaller study (only about 360 subjects) and is potentially confounded by the presence of type 2 diabetic subjects.

Response:

The interest of including the QMDiab study as a second replication cohort is that it shows – despite its weaknesses, that is, multi-ethnic, low power, and being a diabetes case-control study– that the signals we report are robust with respect to moving to other populations, strong enough to be replicated in a lower powered study, and driven by BMI rather than diabetes (as we use diabetes as a covariate, diabetes-driven signals should disappear from the BMI associations).

We clarified this as follows in the paper:

“Despite the fact that QMDiab has a linkage disequilibrium (LD) structure differing from European populations, as well as being multiethnic, diabetes-directed and of limited power, we none the less observe an association, supporting the robustness and strength of the observed signals. As we used diabetes as a covariate, signals are likely driven by BMI rather than diabetes. Thus, while the association between the GPSBMI and BMI in QMDiab compared to the European cohort may have been weaker, the signals we do replicate are likely strong true positives. It remains open to speculation whether generalization of the GPSBMI score was limited due to differences in the genetic architecture of obesity between the populations or due to the sample size.”

Cohorts. The authors state that they tested potential confounding effects and didn't find any, which seems strange given the strong correlation between T2D and obesity. This might reflect a lack of power in the analysis. The potential lack of power is also cited by the authors for several of the analyses.

Response: Obesity and T2D are correlated and could be confounding for certain proteins. However, that does not mean that the specific associations we report in this study must be confounded. For example, as we outline below, we analyzed the WFKINN2 association with BMI, diabetes, and in a combined model (BMI and diabetes in QMDiab). We found that these were two independent signals. This is another motivation for keeping the QMDiab cohort a part of this study.

The first model shows a strong association between WFKINN2 and BMI.

`summary(lm(WFKINN2 ~ bmi + sex + age))`

Coefficients:

	Estimate	Std. Error	t value	Pr(> t)
(Intercept)	0.531038	0.319022	1.665	0.096886 .

bmi	-0.034081	0.009206	-3.702	0.000248 ***
sex	0.258894	0.109026	2.375	0.018103 *
age	0.007551	0.004153	1.818	0.069873 .

The second model shows no significant association between WFKINN2 and diabetes.

`summary(lm(WFKINN2~ diabetes + sex + age))`

Coefficients:

	Estimate	Std. Error	t value	Pr(> t)
(Intercept)	-0.338994	0.223822	-1.515	0.131
diabetes	0.011207	0.122280	0.092	0.927
sex	0.157236	0.107549	1.462	0.145
age	0.005463	0.004824	1.132	0.258

The third model shows that the association between WFKINN2 and BMI is almost unchanged when we account for diabetes as a covariate.

`summary(lm(WFKINN2~ bmi + diabetes + sex + age))`

Coefficients:

	Estimate	Std. Error	t value	Pr(> t)
(Intercept)	0.578149	0.329100	1.757	0.079831 .
bmi	-0.034814	0.009298	-3.744	0.000211 ***
diabetes	0.071667	0.121160	0.592	0.554564
sex	0.262238	0.109273	2.400	0.016923 *
age	0.006201	0.004742	1.308	0.191787

Please also note that, the association between diabetes and BMI in QMDiab is weak ($p=0.00289$). The non-diabetics have a mean BMI of 28.8 while the diabetics have a mean BMI of 30.4.

Cohorts. Secondly, this is a highly admixed population of Middle Eastern and Asian origin (see also comments about genetic risk score estimates). In the end only 20% of the protein associations can be replicated in QMDiab. In this reviewer's opinion, the QMDiab cohort does not contribute significantly to the analyses and conclusions of the study and could be removed.

Response: Please see our response(s) above regarding why we believe QMDiab is still valuable and should be included in this study.

Genetic risk score. The authors used data from the UK biobank to estimate the weights applied to building the PRS. Whereas it can be argued that the German Kora cohort is reasonably genetically similar to use

these weights, this is not the case for the QMdiab study, which is a highly admixed population of Middle Eastern and Asian origin. There is no à priori reason that the weight estimates should apply in this population. An unweighted score (based on simple allele count) could also be computed and results compared.

Response: We agree with the reviewer that there is no à priori reason that the weight estimates should apply in this population. Translation of PRS scores between populations is an ongoing field of research, and while our study may be not powered enough to answer this question in full, we believe that reporting the performance of such scores on new populations is of interest for this question. However, we would not go as far as comparing different scores on a study the size of QMDiab, and therefore did not test alternative ways to derive scores.

Genetic risk score. Also, for the analyses the authors include only the first three genetic PCs. This is less than what is habitually used. The authors should provide a rationale for choosing the first 3 PCs (by providing variance explained for instance) and a plot for the first 2 PCs with different colors for each cohort. This may also be of specific importance for the Middle East population included in this study as some Middle Eastern populations show a high degree of co-sanguinity (up to 30%) that strongly influence the genetic substructure of the population. This is also important in terms of the Mendelian randomization strategy that assumes random mating.

Response: Below we show the explained variance for the genetic PCs for QMDiab. As shown, the majority of variance is explained by the first 3 PCs.

The first three genetic PCs of the QMDiab participants effectively segregated the ethnic groups of the participants as shown in the PC plots below.

Regarding the issue of high degree of co-sanguinity in QMDiab, we are confident there is no significant co-sanguinity in QMDiab because this study is not a Qatari population study. As a matter of fact, there were very few Qataris and the majority of participants were Arabs from a wide range of countries (as can be seen in the spread of the Arab participants in the above PC plot), Indians, or Filipinos. Furthermore, the participants were individually recruited at the dermatology department and are most likely all un-related individuals.

As for KORA, many previous studies have already shown that genetic PCs do not present any potential sources of variation (Please refer to our response to reviewer #1 who raised the same point).

We added the following lines to the methods to address all these points:

“The majority of participants were Arabs, Indians, or Filipinos. The participants were individually recruited (un-related individuals).”

“Together, the first three genetic PCs from QMDiab explained the majority of the genetic variance (13.1%, 5.9%, and 4.0% respectively) and effectively separated the three main ethnic groups composing QMDiab.”

Genetic risk score. It would also be of interest to see the impact of SNPs used in the PC and the genetic score. How many of them are in both components? Removal of genetic score SNPs from the PCs as well as SNPs in LD might improve the association results.

Response:

We checked if any of the first 10 PCs are correlated with the BMI score. We found no such correlation between the BMI score and the 10 first PCs, as shown below:

	Correlation Coefficient	P-Value
PC1	-0.039	0.493
PC2	0.040	0.484
PC3	0.007	0.903
PC4	0.018	0.748
PC5	0.052	0.353
PC6	-0.021	0.705
PC7	0.049	0.390
PC8	-0.052	0.356
PC9	0.050	0.375
PC10	-0.025	0.654

Genotyping. The Kora and QMDiab cohorts were genotyped on different chips (Affymetrix and Illumina respectively). It seems that the authors used different imputed SNP sets for the PRS calculations. Did they create a combined minimal set of imputed SNPs for both cohorts? In the absence of information of the LD structure it is difficult to interpret the GPS-BMI results (do they pertain to the same set of loci or are they different)?

Response:

Over 95% of the variants used in the score calculation were common between KORA and QMDiab, and the correlation (R^2) between scores computed individually and on the intersection was larger than 0.97. We therefore believe that using slightly different SNP sets does not impact the results.

We have added the following details to the Methods section:

“The list of variants comprising the polygenic score for BMI from Khera et al., includes 2,100,302 variants [17]. Imputed genotyping data was used, and in total, 1,583,718 (74.5%) and 1,636,172 variants (77.9%) passed QC for the GPS computation, in KORA and QMDiab respectively. The common set of variants between the two cohorts consisted of 1,565,281 variants (74.5%), and this set included 98.8% of the variants used for the score computation in KORA and 95.7% of the variants used for the score computation in QMDiab. We found the correlation between the score computed using all available variants in KORA and the intersection set to be $R^2=0.99$. In addition, the score computed using all available variants in

QMDiab and the intersection set was $R^2=0.97$. Limiting the score computation to the same set of loci common among the two studies yielded minimal differences in the scores. Therefore, all available variants were included in the score computation for the two studies.”

Functional interpretation of GPS effects. The authors perform analyses to dissect the potential SNP effects (cis- or trans) for the variants in the genetic score and they use a distance metric to remove potential strong cis-effects to evaluate if the overall effects are mostly cis- or trans- driven.

Response:

We checked the distribution of effect weights across the 2 million variants used in the score computation, and found that the majority of variants had a very small effect weight. As shown in the histogram below, a very small number of variants had relatively higher weights compared to the majority of variants. The variants with largest weights were strong BMI QTLs.

As shown in the more detailed weights distribution here, 25% of the variants had an effect weight that fell between 4.72×10^{-12} and 5.66×10^{-6} . On the other hand, 1% of the variants had an effect weight between 1.04×10^{-4} and 6.84×10^{-3} .

Quantile	0%	25%	50%	75%	95%	99%	100%
Effect weight	4.72×10^{-12}	5.66×10^{-6}	1.25×10^{-5}	2.40×10^{-5}	5.64×10^{-5}	1.04×10^{-4}	6.84×10^{-3}

Functional interpretation of GPS effects. They should provide a short summary of the number of potential cis- vs trans effects.

Response:

The contribution of cis and trans effects to the overall signal (based on their weights towards the score) depends on the chosen cutoff for cis versus trans effects as well as the specific gene encoding region for a particular protein. Out of the 1,583,718 variants used for the score computation in KORA, the average contribution of cis-effects within 1MB for the 19 proteins was <1% compared to a contribution >99% for the trans effects. However, if we selected a 100MB cis-window for the 19 proteins, the average contribution of cis-effects became <15%, compared to a contribution of >85% for the trans effects.

Also, it has been shown that cis- SNPs can have effects over large distances due to the 3-dimensional structure of DNA. The metric used (a fixed 1 Mbase interval) is a very crude solution to remove potential cis-acting SNPs.

Response:

The majority of variants had a relatively low effect weight, especially the cis-SNPs (for our proteins of interest). So removing the cis-effects minimally affected the scores (please refer to our previous response).

A LD based approach and/or an approach using publicly available eQTL data for the genetic regions in question would be more appropriate.

Response:

As we previously stressed, there was no large BMI cis effects for any of our proteins of interest (for both the 1 MB and the 100 MB window). Therefore, using an LD based approach would not make a bigger difference.

It seems also strange that the cis- SNPs seem not have a significant effect on the associations, which would mean that most cis-effects are weak and/or that there were not many cis-effect SNPs in the GPS, both of which would be contrary to the existing protein QTL literature (e.g. Chick et al. 2016) and should be discussed. If there are such strong trans- effects it would be worthwhile to investigate and mention and discuss some of the associated gene loci, as they may hold the actual mechanistic clues.

Response:

The reviewer mentioned that they find it “strange that the cis- SNPs seem to not have a significant effect on the associations, which would mean that most cis-effects are weak and/or that there were not many cis-effect SNPs in the GPS, both of which would be contrary to the existing protein QTL literature (e.g. Chick et al. 2016) and should be discussed.” We can argue against this statement because although there were indeed many potential cis-effect SNPs in the GPS, for the proteins of interest, these do not necessarily have strong BMI effects. The only expected strong effects in a BMI score would be for variants strongly associated with BMI, and not necessarily with proteins. For our 19 proteins of interest in this particular study, only the following four proteins had strong known cis/trans pQTLs. Data is extracted from our previous pGWAS server) <https://metabolomics.helmholtz-muenchen.de/pgwas/>)

Protein	Cis/trans pQTL	Variant	P-value
WFIKKN2	cis	rs3803884 17:48,919,039	1.4×10^{-40}
DKK3	cis	rs11022119 11:12,050,652	2.8×10^{-13}
JAG1	trans	rs651007 9:136,153,875	4.7×10^{-11}
C1S	trans	rs6695321 1:196,675,861	9.4×10^{-40}
C1S	trans	rs12734260 1:196,865,417	5.7×10^{-16}

As mentioned in our previous comment, although we removed *cis*-signals within 100 MB and removed known pQTLs (both *cis* and *trans*), only minor changes were observed in the score anyway (See **Supplementary Table 7**).

Tail effect. The authors report a significant tail effect for the distribution of the GPS-BMI score variants and a similar effect for the 19 GPS associated proteins. This is an interesting observation and it could mean that there is a threshold that distinguished “mainly” genetic obesity from “more” environmental obesity. This might also explain some clinical observations that people with higher BMI’s seem to have an earlier onset of obesity than those with overweight/moderate obesity. This might be important for the treatment strategies of these individuals. Do the authors observe a similar tail effect if they use all 152 BMI associated proteins? If not, this would indeed argue for a very strong genetic effect on the extremes of the BMI distribution.

Response:

The tail effect is observed when the ratio of the effect at the tails to the effect of the entire distribution is greater than 1. We originally set our threshold for protein-score associations to a Bonferroni significance level ($p < 0.05/921 = 5.43 \times 10^{-5}$). At this cutoff, only 19 proteins passed stringent significance.

In **Table 2**, we defined our tail effect based on having $P_{full} < 5.43 \times 10^{-5}$, having $P_{5\%} < 0.05$, and most importantly, having at least a 3-fold increase/decrease in effect sizes (beta) of the protein levels between the full data set and limited to data in the 5% tails.

We checked the remaining 133 proteins for a similar tail effect as was observed in these 19 proteins. The statistics were already computed and presented in **Supplementary Table 8** for all 921 proteins in our previous submission. We now extracted the 133 proteins in a new separate sheet (**Supplementary Table 9**) to focus on them.

81 out of 133 proteins were associated with BMI ($p < 5.43 \times 10^{-5}$), but weakly associated with the score ($0.05 \leq p \leq 6.30 \times 10^{-5}$). Of these 81 proteins, 49 proteins were weakly associated with the score at the tails ($p < 0.05$) and had a >3-fold increase/decrease in effect size between the 5% tails and the full data.

On the other hand, 52 out of 133 proteins were associated with BMI ($p < 5.43 \times 10^{-5}$), but not with the BMI score ($p > 0.05$). However, 11 of these 52 proteins were associated with the tails of the score ($p < 0.05$) and had a >3-fold increase/decrease in effect size between the 5% tails and the full data.

We have added the above text describing this extra analysis to the Results and to the Discussion:

“A tail effect is observed when the ratio of the effect at the tails to the effect of the entire distribution is greater than 1.”

“We further tested whether a similar tail effect was observed for the remaining 133 BMI-associated proteins (Supplementary Table 9). 81 out of 133 proteins were associated with BMI ($p < 5.43 \times 10^{-5}$), but weakly associated with GPSBMI ($0.05 \leq p \leq 6.30 \times 10^{-5}$). Of these 81 proteins, 49 proteins were weakly associated with the tails of GPSBMI ($p < 0.05$) and had a greater than 3-fold increase/decrease in effect size between the 5% tails and the full data. On the other hand, 52 out of 133 proteins were associated with BMI, but not at all with GPSBMI ($p > 0.05$). However, 11 of these 52 proteins were associated with the tails of GPSBMI ($p < 0.05$) and had a greater than 3-fold increase/decrease in effect size between the 5% tails and the full data.”

“Common diseases are a result of a complex interplay between genetics and a broad range of environmental perturbations. Exposure to environmental factors (ie. diet, age, exposure to toxins) activates highly interacting protein networks [43], which in turn, may drive molecular mechanisms towards disease. This is likely the case for obesity, where environmental contributions to BMI are well recognized [19]. A tail effect similar to the one previously reported for GPS_{BMI} [17] was observed for the 19 GPS associated proteins and another 49 BMI-associated proteins (at a weaker threshold), but not for 52 proteins that were associated with BMI and not with GPS_{BMI} . The former set of proteins supports the presence of a strong genetic effect on the proteins. Thus, the driving factors for obesity may be distinguished as genetic or other environmental factors. This may aid in explaining clinical observations, such as why a subset of individuals experiences an earlier onset of obesity, and may be useful in defining treatment strategies.”

Proof of causality. Out of 152 BMI associated proteins 30 seem to be causally linked to BMI (24 driven by BMI, 6 driving BMI). Is there any functional connection between these 30 proteins with the 120 proteins for which no effects were observed? In this context also the limitations of the MR in determining causality should be discussed in more detail.

Response:

Formal enrichment analysis using a targeted panel, such a Somalogic’s is challenging due to the bias of which proteins can be measured using the platform.

Regarding the MR, we have added some text to the limitations.

“As with all MR studies, limitations are statistical power, potential reverse causation, population stratification, confounding, and pleiotropy [67]. Although we took precautions to apply only valid MR instruments and report associations at conservative levels of Bonferroni significance, inference of causality should still be interpreted with caution since the validity of MR analyses is based on assumptions and has several limitations as outlined in recent reviews [67-69].”

The term causality as it is used by the authors needs some clarification and should be discussed. For 24 proteins BMI seems to be the driver for the observed increase/decrease of protein concentrations. But BMI itself is a complex, heterogenous phenotype. One example is leptin. Leptin increases with BMI as a result of the expanding fat tissue combined with a peripheral resistance to leptin signaling. BMI and fat mass are obviously associated, explaining the strong correlation with leptin. However, this does not explain the causal molecular mechanism that leads to leptin resistance and the inhibition of the feedback loop that should result in increased satiety signals as fat mass expands. Thus stating that BMI causes high leptin levels does not provide mechanistic insights.

Response:

We explain what causality in MR actually means. Causality in MR is defined as a modification of an exposure that leads to a change in the outcome. Causality in this sense is not indicative of a particular molecular mechanism. Simply, it means if we modify the exposure, we find an effect on the outcome (for example drug target validation etc). If a certain protein is causal for BMI, ie. if we can bring down its level, that suggests that the BMI will go down, but it does not provide any mechanistic insights per se.

This is a caveat and is added to the text:

“Causality in MR is defined as the fact that a modification of an exposure leads to a change in the outcome. Causality in this sense is not indicative of a particular molecular mechanism per se. It simply suggests that modifying the exposure will necessarily lead to a predictable effect on the outcome.”

Discussion. Generally, the discussing lacks in a critical assessment of the potential limitations of the analyses performed and should be reworked in this respect. Although not a major point, the limitations of the Somalogic technology should also be discussed (e.g. the fact that this technology provides relative abundance but not absolute concentrations, potential bias because of the choice of proteins etc.).

Response:

We have added a limitations section to the discussion addressing this:

“Limitations.

We are aware of several limitations to our study. First, the SOMAscan technology provides relative abundance of protein levels, not absolute concentrations. However, this is not a concern for association studies. Second, the findings reported here are limited to the specific proteins included in the SOMAscan panel, and also to proteins associations that are detectable in blood. Therefore, the list of associations we report here is not comprehensive, and studies using other technologies and other biological sample types may reveal further associations. The disease areas, physiological processes, and classes of the proteins included in the SOMAscan assay have been described in our previous study [65]. Third, aptamer-based proteomics methods are sensitive to potential probe cross-reactivity and non-specific binding. We include the validation information for all proteins extracted from two studies in Supplementary Table 2. A full review of the limitations of SOMAscan technology such as possible epitope effects (influence of genetic variance), unspecific binding, cross-reactivity, interference with DNA-binding proteins, limited coverage of isoforms and protein post-translational modifications has been described elsewhere [66].”

Discussion.

The focus of the leptin and leptin receptor animal models in the discussion seems strange, given that the authors have found some potentially interesting new results for some of the proteins for which there are mouse ko. Indeed, some of the text in the results section is more descriptive of these animal findings and should be moved to the discussion and be extended.

Response:

We felt that before focusing on new results we should mention LEP/LEPR animal models since it supports the rationale of our study. But now we equally focus on the potentially new and interesting hits along with LEP and LEPR. We moved the descriptive parts of the animal findings to the discussion and extended. Please see tracked change document for changes.

Discussion.

The discussion about the potential drug targets is highly speculative as there is no evidence that these targets can be used in obesity treatment from the available data and should be either shortened or removed.

Response: We prefer keeping the drug targets part over removing it, but we now shortened it. Causality in MR implies that a change of an exposure will lead to a change in the outcome. After appropriate consideration of all caveats to MR, our MR analysis indeed provides evidence that these drug targets are expected to change BMI since they change the protein levels. Such MR evidence may be of interest from a pharmaceutical perspective.

Reviewer #3 (Remarks to the Author):

The paper presents a series of analyses from three cohorts to examine the relationship between plasma protein levels and obesity, as measured by body mass index (BMI). The work first establishes association between protein levels and BMI, then tested these proteins for association with a genome-wide polygenic risk score for BMI developed from published data. This was then followed by bi-directional MR analysis to infer causal relationships between protein levels and obesity. Finally, through bioinformatics and literature search, the authors posit potential functional roles for implicated/causal proteins. The paper asks a series of clear and understandable questions about circulating protein levels and their impact on/from obesity. While the sets of analyses make good sense and address appropriate questions for these data, several aspects need to be clarified before publication.

Response: We thank the reviewer for their interest in our study. We addressed the specific concerns as described below. We hope we answered all questions and provided satisfactory clarification where needed.

RE: Relation of associated proteins to GWAS/eQTL loci.

1) How many proteins fall in known BMI loci or have eQTLs near BMI loci

Response: We used data from Phenoscanner to query which of the 152 BMI-associated proteins were associated with known BMI loci or are eQTLs near BMI loci.

We added the following text to the results:

“We then queried Phenoscanner [28] [29] to determine which of the 152 BMI-associated proteins were associated with known BMI loci or may be considered the best candidate in the genomic vicinity. We found that five proteins were strong pQTLs for BMI loci/regions. These included Leptin (LEP), C-reactive protein (CRP), Apolipoprotein B (APOB), Lysosomal protective protein (CTSA), and Neural cell adhesion molecule 1, 120 kDa isoform (NCAM1). Further proteins were found to represent eQTLs near BMI loci, including Immunoglobulin M (IGJ), Interleukin-1 Receptor accessory protein (IL1RAP), Calpastatin (CAST), Apolipoprotein B (APOB), Platelet-activating factor acetylhydrolase (PLA2G7), Plasma protease C1 inhibitor (SERPING1), Reticulon-4 receptor (RTN4R), Insulin-like growth factor 1 receptor (IGF1R), Integrin alpha-V: beta-5 complex (ITGB5), Complement factor B (CFB), Complement component 1 Q subcomponent-binding protein, mitochondrial (C1QBP), Cell adhesion molecule 1 (CADM1), Galectin-3-binding protein (LGALS3BP), and Antithrombin-III (SERPINC1).”

RE: Relation of associated proteins to GWAS/eQTL loci.

2) Could be/have been considered best candidate in the regions (closest gene, etc.)

Response:

We used data from Phenoscanner to query which of the 152 BMI-associated proteins can be considered the best candidate in the region of known BMI loci. We combined this answer in the response above.

RE: Relation of associated proteins to GWAS/eQTL loci.

3) Does this data help inform on "causal" loci in GWAS regions? Provide new insights?

Response:

Finding relationships between associated proteins and GWAS/eQTL loci can certainly emphasize potential causal loci. However, MR is still needed to determine causal relationships. The following was added to the discussion:

“The GPS_{BMI} not only captures strong BMI variants, but genome-wide BMI effects (although the former would have stronger weights). The overlap we find between BMI loci and pQTLs/eQTLs ie. LEP, CRP, NCAM1, CTSA, LGALS3BP, IGJ, and SERPINC1 provide useful insight for causation. We later used MR to distinguish between the contribution/consequence of these proteins with respect to BMI.”

RE: Protein analysis

1) Are there specific kinds of proteins most/least amenable to analysis with this method? How would this impact the ability to discover associations, particularly for BMI.

Response:

Details of the kinds of proteins used in this study were already described in our previous study [64]. The aptamers on the SOMAlogic panel we used have been generated to cover numerous disease areas and physiological processes. The panel also targets various classes of proteins, covering secreted, intracellular and extracellular domains of proteins.

We have added a reference to these details in the limitation section at the end of the discussion. Please see previous comment.

RE: Protein analysis

2) Relevance/generalizability of plasma protein levels for obesity generally and vis a vis other obesity-related tissues.

Response:

Regarding the relevance or generalizability of plasma for obesity and obesity related tissues, we have added the following text to the discussion:

“Blood circulating proteins permeate the entire organism and may be involved in direct regulation of complex diseases such as obesity or diabetes. Protein associations may provide biological interpretations of the molecular mechanisms occurring due to increased BMI and obesity.”

RE: MR analyses

1) MR assumes that genetic variants have no other effect on the outcome of interest (either BMI or proteins) except through the outcome risk factor. As described, I'm not convinced that either the BMI GPS or the individual protein GPS's satisfy this assumption. If they do, please clarify in the methods how the instruments were derived to satisfy any effects not through the risk factor. If they do not satisfy this assumption, analysis should be redone to account for this assumption.

Response:

1) *We carried out the MR following the widely applied and best known practices to assess the issue of heterogeneity or horizontal pleiotropy. Given the known potential limitations of MR, we used the MR-Egger method for evaluation. We strengthen our inference using the MR Egger method and details are described in **Supplementary Table 10**. Briefly, to assess the causal estimate of BMI on the proteins, for which data was available, we initially used the inverse-variance weighted method (IVW). We compared the results from the IVW method to the MR Egger method. The estimate from the Egger method was consistent with the IVW estimate for all tested proteins. The Cochran's Q statistic (I^2 , $sq.egger$) represents the heterogeneity (a measure of the degree to which genetic instruments identify the same causal effect). The significance of this heterogeneity is represented by p_{Het} and a low p-value corresponds to evidence of non-directional pleiotropy. P_{pleio} is the p-value for the MR-Egger intercept test and a low p-value corresponds to evidence of directional pleiotropy. We did not find any evidence of non-directional nor directional pleiotropy for any of the tested proteins (i.e. $p_{Het} > 0.05$ and $p_{pleio} > 0.05$ for all tested proteins. We have also clarified this in the methods section as follows:*

“These results were also robust to sensitivity analysis and evidence of heterogeneity or horizontal pleiotropy, based on the MR Egger analysis, was weak (Supplementary Table 10). For all tested proteins, the heterogeneity measures represented by the Cochran’s Q statistic were not significant ($p_{\text{Het}} > 0.05$), suggesting there was no non-directional pleiotropy. In addition, we did not find any evidence of directional pleiotropy, according to the MR-Egger intercept test ($p_{\text{Pleio}} > 0.05$).”

RE: MR analyses

2) MR also assumes there are no confounders of the genetic variant-outcome association, but no attempt is made to adequately assure that this assumption is also satisfied.

Response:

2) *To the best of our knowledge, we have already tested for numerous potential confounders of the genetic variant-outcome associations. This is now better clarified in the Methods as follows:*

“We further eliminated SNPs with potential confounders using data from the UK Biobank GWAS [82]. We downloaded all variant association data with potential confounders (education, smoking, alcohol use, physical activity etc.) from the UK Biobank GWAS (determined by genome-wide significance of $p < 1 \times 10^{-8}$). After these filtration steps and elimination of potentially confounded SNPs, 454 BMI instruments were used.”

RE: Figure 3

1) The X axis reflects deciles (0-10, 11-20, etc.), not percentiles, correct?

b) It seems odd to see essentially no effect on LEP levels outside of top and bottom deciles.

c) Does the GPS include SNPs associated with LEP levels?

Response:

1) *We changed the X-axis to deciles instead of percentiles.*

b) *Although **Figure 3b** seems to show no effect on LEP outside the top and bottom deciles, we do observe an effect on LEP levels when looking at the various BMI extremes (**Figure 4c**). It is difficult to answer this question, but we could speculate that this might be due to environmental effect(s) as the previous reviewer suggested. For instance, LEP is known to have a strong gender effect.*

c) *The GPS does not include SNPs associated with LEP levels. Please refer to previous comment.*

Given the comments on environmental contributions to BMI, it might be useful to comment on the lack of effect of "environmental" risk factors on the protein/BMI associations?

Response: We agree with the reviewers that contributions to BMI can be both genetic and environmental. Please see our previous response to the tail effect comment. We have also added the following text to the discussion:

“Common diseases are a result of a complex interplay between genetics and a broad range of environmental perturbations. Exposure to environmental factors (ie. diet, age, exposure to toxins) activates highly interacting protein networks [43], which in turn, may drive molecular mechanisms towards disease. This is likely the case for obesity, where environmental contributions to BMI are well recognized [19].”

We also noted previously:

“We tested the influence of a number of potential confounders on the BMI-protein associations, specifically, smoking status, alcohol consumption, physical activity, and diabetes state (Supplementary Table 3). We did not observe any substantial effect of confounding by these factors on the BMI-protein associations, and all Bonferroni significant protein-BMI associations found in the full model were also significant in the model where only age and sex were included as covariates.”

Similarly, there is no discussion of the relevance of the broadly expressed versus sporadically expressed tissues and what importance to place on these results. Could pathway analyses be useful?

Response:

The importance of this observation is simply to mention that some proteins have global functional roles while others are geared toward more specific pathways. We clarified this by adding the following text:

“While the functional role of a subset of these may be more global, others may imply specific pathways.”

Regarding the pathway analysis, please refer to our previous response to Reviewer #2. We also mentioned the limitation of SOMAlogic technology in the caveat section in the manuscript.

Given a known and strong role for the CNS in BMI/obesity biology, why were comparisons only done in adipose and liver tissue for follow-up studies. Why not consider roles for CNS tissues too?

Response:

*We agree with the reviewer that it is important to carry out comparisons in other tissues, since obesity does not only act through the liver or muscle. So we added a mouse expression follow-up study for brain tissue to **Figure 6E**.*

We added some text to the manuscript.

“In general, correlations between proteins were similar, but stronger in adipose tissue compared to liver and brain tissues. However, a number of differences are noteworthy. For instance, WFIKKN2 was associated with triglycerides and lipids in adipose and brain tissue, but not liver. The association of WFIKKN2 and triglycerides and total fat in brain tissue was previously reported [27].”

The authors offer competing explanations for diminished performance in the QMDiab study, poor performance of European GPS and power. Both are plausible, but no effort is made to distinguish between the two possibilities? Can this be determined empirically?

Response: Please refer to our response to Reviewer # 1 who suggested removing QMDiab from this study. The purpose of including QMDiab in our study is to show the robustness of the observed signals when moving to other populations. However, it would not be sufficiently powered to address this question empirically.

“Despite the fact that QMDiab has a linkage disequilibrium (LD) structure differing from European populations, as well as being multiethnic, diabetes-directed and of limited power, we none the less observe an association, supporting the robustness and strength of the observed signals. As we used diabetes as a covariate, signals are likely driven by BMI rather than diabetes. Thus, while the association between the GPSBMI and BMI in QMDiab compared to the European cohort may have been weaker, the signals we do replicate are likely strong true positives. It remains open to speculation whether generalization of the GPSBMI score was limited due to differences in the genetic architecture of obesity between the populations or due to the sample size.”

Reviewer #1 (Remarks to the Author):

The authors fully addressed my comments

Reviewer #2 (Remarks to the Author):

General remarks

The authors have made a considerable effort to respond to the questions and concerns raised. Most of my concerns/questions have been adequately addressed by the authors and so have the remarks from the other reviewers as far as I can tell.

Overall I think that the particular strengths of this work lie in the approach to couple the genetic analysis to the proteome, which allows to identify potential protein drivers of obesity. The observation of the disproportional genetic effect on the BMI tails is also very interesting and important.

I have a couple of minor comments concerning the responses to questions raised by me and reviewer 3 about the QMDiab cohort and some functional or pathway analysis for the identified causal proteins in the MR analysis:

1- QMDiab study

The authors respond that they still believe the results should be shown as "...that the signals we report are robust with respect to moving to other populations..." However, only 20% of the signals replicate statistically significantly. In their discussion they conclude: "...It remains open to speculation whether generalization of the GPSBMI score was limited due to differences in the genetic architecture of obesity between the populations or due to the sample size." However, the whole point of a replication cohort for the genetic score is to obtain evidence for this generalization. Whereas leaving the results of QMDiab in the article does not do any harm to the overall message of the paper I maintain that it also does not add any particular further insights.

2- Pathway analysis

Both this reviewer as well as reviewer 3 ask for some kind of "pathway" analysis for the identified proteins. The authors reply that formal enrichment analysis would not be possible (or rather would be highly biased) due to the fact that the Somalogic protein array is neither comprehensive nor random in its content. I appreciate this fact and this is not what I asked for.

Rather it would be interesting to get some idea of molecular, biological context, especially for the proteins identified as potentially causal. This kind of analysis can be carried out in silico and does not depend on the Somalogic content. Three of the six potentially causal proteins (AGER, DPT, and CTSA) are new and more biological context (e.g protein-protein interaction network) might provide valuable clues for future mechanistic studies.

Finally, after re-reading the article I think that the title does not describe the work adequately as rather than elucidating the genetic risk of obesity, the paper elucidates the behaviour and role of the proteome in respect to genetic drivers or by using MR to dis-entangle the causal directionality of the protein changes. Maybe they should consider to change the title to reflect this.

Reviewer #3 (Remarks to the Author):

The authors have worked hard to address the many comments and questions of each of the reviewers and the paper is a compelling addition to the biological understanding of obesity. My only outstanding comment is that I'd prefer a more detailed discussion of the impact of the selected proteins targeted/analyzed on the results of the study. While I agree that pathway analyses may not be appropriate, due to the selected nature of the proteins, at the same time, I worry that substantial bias in the kinds of proteins selected by the SomaLogic technology might have unexpected or undesirable consequences.

Point-by-point response to the referees' comments

"Elucidating the genetic risk of obesity through the human blood plasma proteome"

Zaghlool et al.

Reviewer #1 (Remarks to the Author):

The authors fully addressed my comments.

Response: We thank the reviewer for their interest in our study.

Reviewer #2 (General Remarks):

The authors have made a considerable effort to respond to the questions and concerns raised. Most of my concerns/questions have been adequately addressed by the authors and so have the remarks from the other reviewers as far as I can tell.

Overall I think that the particular strengths of this work lie in the approach to couple the genetic analysis to the proteome, which allows to identify potential protein drivers of obesity. The observation of the disproportional genetic effect on the BMI tails is also very interesting and important.

I have a couple of minor comments concerning the responses to questions raised by me and reviewer 3 about the QMDiab cohort and some functional or pathway analysis for the identified causal proteins in the MR analysis:

Response: We thank the reviewer for their interest in our study. We addressed the specific concerns as described below. We hope we answered all questions and provided satisfactory clarification where needed.

1- QMDiab study

The authors respond that they still believe the results should be shown as "...that the signals we report are robust with respect to moving to other populations..." However, only 20% of the signals replicate statistically significantly. In their discussion they conclude:...It remains open to speculation whether generalization of the GPSBMI score was limited due to differences in the genetic architecture of obesity between the populations or due to the sample size."

However, the whole point of a replication cohort for the genetic score is to obtain evidence for this generalization. Whereas leaving the results of QMDiab in the article does not do any harm to the overall message of the paper I maintain that it also does not add any particular further insights.

Response: We agree that QMDiab is not a full-fledged replication cohort, and as such, non-replication of an association signal cannot be interpreted as a false positive hit. However, in cases where replication is successful, it indicates that these signals are particularly robust and worthy to be followed up-on with priority, as they are strong enough to be seen even under less ideal conditions and in different populations.

In terms of generalization of the genetic score, we actually replicated in QMDiab more than half of the 19 proteins associations with the GPS_{BMI} and believe this worthy of mentioning. We therefore feel that the QMDiab part should be kept in the manuscript.

2- Pathway analysis

Both this reviewer as well as reviewer 3 ask for some kind of "pathway" analysis for the identified proteins. The authors reply that formal enrichment analysis would not be possible (or rather would be highly biased) due to the fact that the Somalogic protein array is neither comprehensive nor random in its content. I appreciate this fact and this is not what I asked for.

Rather it would be interesting to get some idea of molecular, biological context, especially for the proteins identified as potentially causal. This kind of analysis can be carried out in silico and does not depend on the Somalogic content. Three of the six potentially causal proteins (AGER, DPT, and CTSA) are new and more biological context (e.g protein-protein interaction network) might provide valuable clues for future mechanistic studies.

Response: We agree with the reviewer that adding some biological context for the new causal proteins might provide valuable clues for future mechanistic studies. However, we feel we already incorporated extensive animal model information in the paper and in Supplementary Data 14 as well as enrichment in specific obesity traits using two different datasets (including adipose, liver, and brain tissue) in Figure 6. Adding protein-protein interaction networks would be out of the scope of the current study. We rather expanded on the enrichment results by referring to the new potentially causal proteins (AGER, DPT, and CTSA) by adding the following text (from data which was already presented in Figure 6).

"We observed enrichment in obesity traits in mice, such as weight, length, and triglycerides for the potentially causal proteins (Ager, Ctsa, and Dpt). We also found enrichment in HDL cholesterol and total cholesterol for Ager and Ctsa, fat mass for Ctsa and Dpt, and abdominal fat for Dpt."

Finally, after re-reading the article I think that the title does not describe the work adequately as rather than elucidating the genetic risk of obesity, the paper elucidates the behaviour and role of the proteome in respect to genetic drivers or by using MR to dis-entagle the causal directionality of the protein changes. Maybe they should consider to change the title to reflect this.

Response: We thank the reviewer for their suggestion and changed the title

accordingly to:

"Revealing the role of the human blood plasma proteome in obesity using genetic drivers"

Reviewer #3 (Remarks to the Author):

The authors have worked hard to address the many comments and questions of each of the reviewers and the paper is a compelling addition to the biological understanding of obesity. My only outstanding comment is that I'd prefer a more detailed discussion of the impact of the selected proteins targeted/analyzed on the results of the study. While I agree that pathway analyses may not be appropriate, due to the selected nature of the proteins, at the same time, I worry that substantial bias in the kinds of proteins selected by the SomaLogic technology might have unexpected or undesirable consequences.

Response: We added the following paragraph as a more detailed discussion of the impact of the selected proteins targeted/analyzed on the results of the study:

"[...] Second, the findings reported here are limited to the specific protein set targeted by the SOMAscan panel, and also to protein associations that are detectable in blood. Therefore, the list of associations we report here is not comprehensive, and studies using other technologies and other biological sample types may reveal further associations. The specific disease areas, physiological processes, and classes of proteins targeted by the SOMAscan assay have been described in our previous study [66]."